# Neutralization of SARS-CoV-2 Omicron BQ.1, BQ.1.1 and XBB.1 variants following SARS-CoV-2 infection or vaccination in children

Lorenza Bellusci[1], Gabrielle Grubbs [1], Shaimaa Sait[1], Lael M. Yonker [2], Adrienne G. Randolph [3,4], Tanya Novak [3,4], Takuma Kobayashi[4], Overcoming COVID–19 Investigators* & Surender Khurana [1]✉

Emergence of highly transmissible Omicron subvariants led to increased SARS-CoV-2 infection and disease in children. However, minimal knowledge exists regarding the neutralization capacity against circulating Omicron BA.4/BA.5, BA.2.75, BQ.1, BQ.1.1 and XBB.1 subvariants following SARS-CoV-2 vaccination in children versus during acute or convalescent COVID-19, or versus multisystem inflammatory syndrome (MIS-C). Here, we evaluate virus-neutralizing capacity against SARS-CoV-2 variants in 151 age-stratified children ( <5, 5–11, 12–21 years old) hospitalized with acute severe COVID-19 or MIS-C or convalescent mild (outpatient) infection compared with 62 age-stratified vaccinated children. An age-associated effect on neutralizing antibodies is observed against SARS-CoV-2 following acute COVID-19 or vaccination. The primary series BNT162b2 mRNA vaccinated adolescents show higher vaccine-homologous WA-1 neutralizing titers compared with <12 years vaccinated children. Post-infection antibodies did not neutralize BQ.1, BQ.1.1 and XBB.1 subvariants. In contrast, monovalent mRNA vaccination induces more cross-neutralizing antibodies in young children <5 years against BQ.1, BQ.1.1 and XBB.1 variants compared with ≥5 years old children. Our study demonstrates that in children, infection and monovalent vaccination-induced neutralization activity is low against BQ.1, BQ.1.1 and XBB.1 variants. These findings suggest a need for improved SARS-CoV-2 vaccines to induce durable, more cross-reactive neutralizing antibodies to provide effective protection against emerging variants in children.

The SARS-CoV-2 Omicron variants continue to evolve, generating multiple sub-lineages with increased transmissibility and antibody-escape mutations resulting in widespread circulation of COVID-19 around the globe[1]. In children, SARS-CoV-2 infection is often asymptomatic or causes mild disease; however, children are susceptible to develop severe manifestations of COVID-19 and its associated post-infectious severe complication Multisystem Inflammatory Syndrome in Children (MIS-C). Several lineages of Omicron that are currently circulating, with predominance of BA.4, BA.5, BA.2.75, BQ.1, BQ.1.1 and recombinant XBB.1, contain key mutations in the receptor-binding

[1]Division of Viral Products, Center for Biologics Evaluation and Research (CBER), FDA, Silver Spring, MD 20993, USA. [2]Mucosal Immunology and Biology Research Center, Massachusetts General Hospital for Children, Harvard Medical School, Boston, MA 02114, USA. [3]Department of Anesthesia, Harvard Medical School, Boston, MA, USA. [4]Department of Anesthesiology, Critical Care and Pain Medicine, Boston Children's Hospital, Boston, MA, USA. [5]A full list of members and their affiliations appears in the Supplementary Information. *A list of authors and their affiliations appears at the end of the paper.
✉e-mail: Surender.Khurana@fda.hhs.gov

domain (RBD) with over 36 mutations in spike protein compared with ancestral WA-1/2020 strain (Table S1). XBB is a recombinant variant, with its genome consisting of a combination of two different 'parent' variants- Omicron BA.2.10.1.1 and BA.2.75.3.1.1.1 lineages, with a breakpoint in the S1 region of the Spike protein. Importantly, these mutations resulted in resistance of these circulating Omicron variants to most therapeutic monoclonal antibodies available for treatment of

COVID-19 as well as escape from vaccination-induced antibodies generated following either parental mRNA vaccine or the bivalent booster in adults[2–6].

In the United States, as of May 10, 2023, vaccination rates in children remain very low, with only 6% of children <5 years of age have completed a primary series of vaccination, and only 8% of children under 18 years of age have received a vaccine booster dose (Fig. 1a and

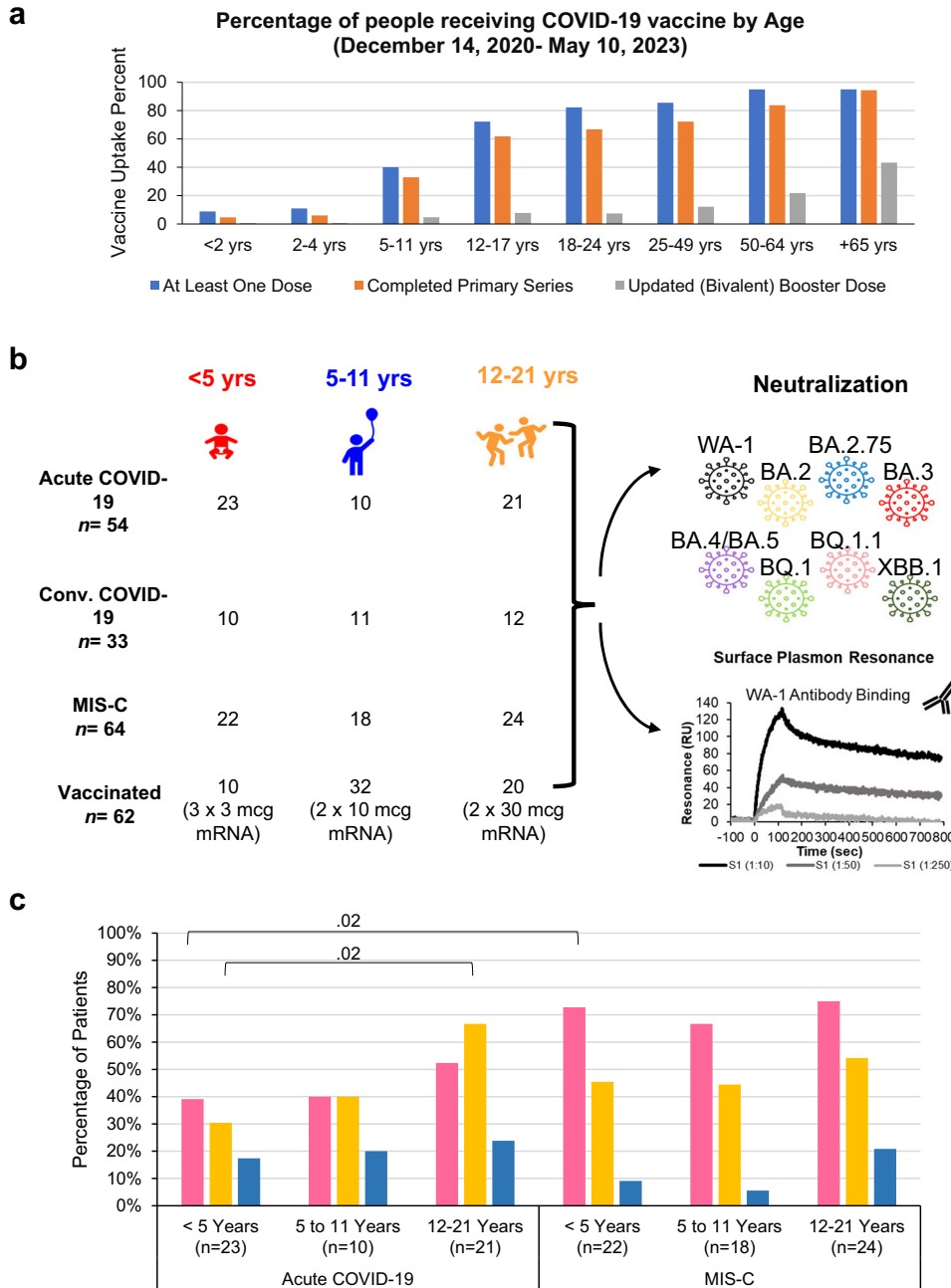

Fig. 1 | Study design of children following vaccination or with acute COVID-19 or convalescent COVID-19 or MIS-C. a Percentage of people who received a single dose, completed primary series or booster dose of COVID-19 vaccine by different age groups in US (from December 14, 2020, to May 10, 2023), as per data from US CDC (https://covid.cdc.gov/covid-data-tracker/#vaccination-demographics-trends). b Overview of children cohort with acute COVID-19 or convalescent COVID-19 or MIS-C or following mRNA vaccination. Each children sample was evaluated for neutralizing antibodies against eight SARS-CoV-2 strains in pseudovirus neutralization assay and for binding antibodies against prototype vaccine-homologous WA-1 RBD using surface plasmon resonance. c Percent distribution of hospitalized acute COVID-19 and MIS-C patients admitted to the intensive care unit (ICU), requiring any respiratory support, and receiving mechanical ventilation. Categorical (yes/no) data were collected from each hospitalized children for disease severity and therefore the frequency of "yes" for each parameter is shown. Statistical differences between age groups within each disease category or between different disease category within each age group were analyzed by Pearson's chi-squared test (comparing 2 age groups at a time) and the statistically significant p-values are shown. The tests were two-sided tests.

https://covid.cdc.gov/covid-data-tracker/#vaccination-demographics-trends). The vaccination rates in children with prior SARS-CoV-2 infection are even lower[7,8]. Moreover, different dosages of mRNA vaccines are given to children in different age groups, which may impact the SARS-CoV-2 immune response generated following vaccination[9]. For the Pfizer BNT162b2 vaccine, the primary vaccination series in younger children <5 years consists of three doses of 3 mcg mRNA vaccine, while children aged 5–11 years get 2 doses of 10 mcg mRNA vaccine, and adolescents (12–21 years) receive 2 doses of 30 mcg mRNA vaccine dosage.

However, limited knowledge exists regarding vaccine-induced antibody responses following primary series of mRNA vaccination in different age groups of children compared with post-infection COVID-19 or post-infectious multisystem inflammatory syndrome in children (MIS-C), and its ability to neutralize circulating highly contagious SARS-CoV-2 variants. It is possible that vaccinated children or children infected with SARS-CoV-2 prior to circulation of Omicron strain remain susceptible to re-infection by Omicron lineages as was observed when Omicron BA.1 emerged in late 2021[10,11]. Thus, they have the potential to transmit it virus to other children and vulnerable populations[12]. Therefore, it is critical to understand the capacity of the antibody response induced by SARS-CoV-2 vaccination compared with post-infection immune response in children of different age groups to neutralize currently circulating highly transmissible SARS-CoV-2 Omicron BA.4, BA.5, BA.2.75, BQ.1, BQ.1.1 and XBB.1 subvariants.

In this study, we evaluated the capacity of neutralizing antibodies induced following primary series of mRNA-based (Pfizer-BioNTech BNT162b2) vaccination compared with post-infection-induced antibodies in children with either acute COVID-19, convalescent COVID-19, or MIS-C, stratified by age <5 years, 5–11 years, and 12–21 years against SARS-CoV-2 ancestral vaccine-homologous WA-1 strain and circulating Omicron BA.2, BA.2.75, BA.3, BA.4/BA.5, BQ.1, BQ.1.1 and XBB.1 variants.

## Results

### Study cohort

Most children in US have not received a SARS-CoV-2 booster vaccine (Fig. 1a). The main aim of the study was to evaluate the capacity of the pediatric antibody response induced by the first generation monovalent COVID-19 mRNA vaccines encoding prototype WA-1 spike antigen compared with those generated following primary SARS-CoV-2 infection by ancestral WA-1 like strains early during the pandemic to neutralize currently circulating highly transmissible Omicron subvariants, in age-stratified children, during acute infection and after convalescence. Therefore, we analyzed post-SARS-CoV-2 infection or post-vaccination samples from a total of 213 US children and adolescents (Fig. 1b). The demographics of the study cohorts are summarized in Table S2. Our hospitalized cohort comprised of 118 acutely ill children, consisting of 64 with MIS-C and 54 with acute COVID-19; and a separate outpatient cohort of 33 children that were ≥30 days post-acute mild COVID-19 (convalescent). Acute samples (serum or plasma) were collected as early as possible during hospitalization and within 2 weeks of presentation. All pediatric patients were confirmed to be SARS-CoV-2 infected either by reverse transcriptase quantitative PCR (RT-qPCR) and/or positive SARS-CoV-2 antibody tests. All post-infection samples were obtained during April 2020 to March 2021, when the prevalent circulating SARS-CoV-2 strains were the WA-1 or the Alpha strain in the United States and were described in a previous study[13]. There was no statistical difference in the timing of pediatric sample collection relative to hospital admission or PCR/antibody positive test between the various age groups in any of the disease cohorts (Table S2).

MIS-C is a hyperinflammatory syndrome that occurs approximately 3–6 weeks post-SARS-CoV-2 infection in children who were asymptomatic or had mild illness upon initial infection. As a control group for MIS-C, we collected convalescent COVID-19 samples from outpatient children and adolescents with asymptomatic or mild illness (never hospitalized) approximately 3–6 weeks after their positive SARS-CoV-2 PCR test. Because MIS-C is rare (~2–3 per 10,000 SARS-CoV-2 infections in US), and it is not possible to identify patients in advance, therefore, it is not feasible to get baseline (during initial infection) samples in most MIS-C patients.

Categorical (yes/no) data were collected from each hospitalized child for disease severity assessment including ICU admission, respiratory support, and mechanical ventilation. Comparing disease severity between acute COVID-19 vs MIS-C across the age groups (Table S2), the children hospitalized with MIS-C had significantly more (p = 0.003) ICU admissions than children with acute COVID-19 (Fig. 1c). Children <5 years old with acute COVID-19 also required significantly less respiratory support than adolescents with the same acute illness (p = 0.02). Among different MIS-C age groups, no significant differences were observed in disease severity based on ICU admission, respiratory support, or mechanical ventilation. None of these children were vaccinated.

The vaccination cohort consisted of 62 SARS-CoV-2 naïve children vaccinated with Pfizer's BNT162b2 mRNA vaccine and sera were collected at three weeks (median 18 days) after completion the primary vaccination series (Table S2). For BNT162b2 vaccine, the youngest children (<5 years) received three doses of 3 mcg mRNA vaccine, while children aged 5–11 years got 2 doses of 10 mcg mRNA vaccine, and adolescents (12–21 years) received 2 doses of 30 mcg mRNA vaccine at three weeks apart.

### Neutralization capacity of post-infection and post-vaccination samples against circulating SARS-CoV-2 variants in children

Pediatric samples were divided by age categories: <5 years (n = 65; 23 acute, 10 convalescent, 22 MIS-C and 10 naïve vaccinated), 5–11 years (n = 71; 10 acute, 11 convalescent, 18 MIS-C and 32 naïve vaccinated) and 12–21 years old (n = 77; 21 acute, 12 convalescent, 24 MIS-C and 20 naïve vaccinated) for evaluation of the antibody response (Fig. 1b). Pseudovirion neutralization assay (PsVNA) was used to evaluate the antibody neutralization activity of the children and adolescents against the ancestral infecting and the vaccine-homologous SARS-CoV-2 WA-1 strain, and the circulating highly transmissible Omicron sublineages in US: BA.2, BA.2.75, BA.4, BA.5, BQ.1, BQ.1.1 and XBB.1. The mutations in the spike protein of these Omicron subvariants compared with vaccine-homologous WA-1 are shown in Table S1. SARS-CoV-2 neutralizing activity measured by PsVNA correlates with PRNT (plaque reduction neutralization test with authentic SARS-CoV-2 virus) in previous studies[14,15].

For the post-infection cohort, among pediatric acute COVID-19, a steady increase in titers of neutralizing antibodies against the ancestral WA-1 infecting strain was observed as their age increased. The adolescents (12–21 years) demonstrated higher neutralization titers (PsVNA50 titers; sample dilution that resulted in 50% virus neutralization) ranging between 10–12,950 [geometric means titer (GMT) of 256] that were 3.3 to 6.9-fold higher compared with infected 5–11 years old (GMT:77; range 10–1114) and infected younger (<5 years) acute COVID-19 patients (GMT:37; range 10–2865), respectively (Fig. 2a). However, samples from children with convalescent non-hospitalized outpatient COVID-19 or inpatient hospitalized MIS-C, demonstrated higher WA-1 neutralizing antibodies in younger children such that PsVNA50 titers were similar across all age groups (Fig. 2b, c). In the youngest children (<5 years), neutralization titers against WA-1 for the MIS-C patients (GMT:576; range 10–5613) or the convalescent COVID-19 (GMT: 479; range 21–1231) samples were 15.6 and 12.9 fold higher than corresponding acute COVID-19 patients (GMT:37), respectively (Fig. S1a). In the youngest age cohort, the seropositivity rate against WA-1 (PsVNA50 titer of >1:40) predicted to provide protection against severe COVID-19[16] increased from 30% for acute

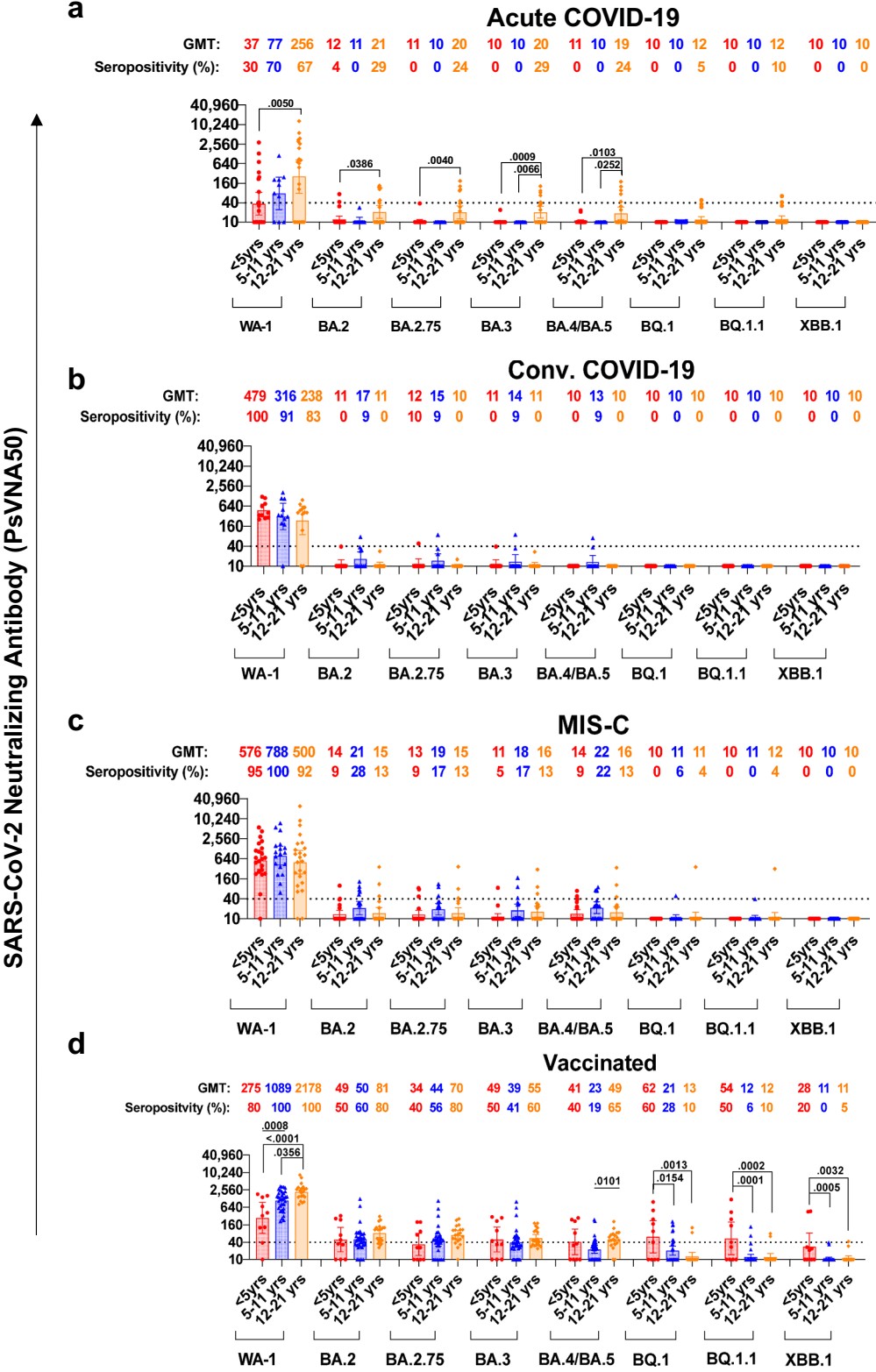

COVID-19 patients to 95–100% for the convalescent COVID-19 children and the MIS-C patients (Fig. S1a). However, the caveat to this correlate of protection that it was defined in the context of vaccination and may not apply to all SARS-CoV-2 variants. For the older children (5–21 years), the anti-WA-1 neutralization titers for MIS-C patients were higher than acute and convalescent COVID-19 patients (Fig. S1b, c).

SARS-CoV-2 spike mRNA (Pfizer BNT162b2) vaccination of naïve healthy children after 28-days of completed primary vaccination schedule induced 2-fold higher neutralization titers in adolescents (GMT:2178; range 807–8407) compared with vaccinated 5–11 years old (GMT:1089; 210-3340) and 7.9-fold higher than the youngest vaccinated children <5 years (GMT:275; range 10–1825) against the vaccine-homologous WA-1, suggesting a vaccine dosage-dependent or age-stratified impact on vaccine-induced antibody response (Fig. 2d). Vaccination induced significantly higher anti-WA-1 neutralizing titers than all three post-infection cohorts in children older than 5 years

**Fig. 2 | Neutralizing antibody titers of serum/plasma from children with COVID-19 or MIS-C or vaccinated children against SARS-CoV-2 variants.** SARS-CoV-2 neutralization assays were performed by using pseudoviruses expressing the spike protein of WA-1 or the Omicron subvariants BA.2, BA.2.75, BA.3, BA.4/BA.5, BQ.1, BQ.1.1 and XBB.1 subvariants, in 293-ACE2-TMPRSS2 cells. **a–d** SARS-CoV-2 neutralization titers were determined in each of the 213 children with either acute COVID-19 (**a**), convalescent COVID-19 (**b**), MIS-C (**c**) or following vaccination (**d**), divided by age categories: <5 years (*n* = 65; 23 acute, 10 convalescent, 22 MIS-C and 10 naïve vaccinated), 5–11 years (*n* = 71; 10 acute, 11 convalescent, 18 MIS-C and 32 naïve vaccinated) and 12–21 years old (*n* = 77; 21 acute, 12 convalescent, 24 MIS-C and 20 naïve vaccinated). The assay of each sample was performed in duplicate to determine the 50% neutralization titer (PsVNA50). PsVNA50 (50% neutralization titer) titers for youngest children (<5 years, shown in red), 5–11 years (shown in blue) and adolescents (12–21 years, shown in orange). The heights of the bars and the numbers over the bars indicate the geometric mean titers (GMT), and the whiskers indicate 95% confidence intervals. The horizontal dashed line indicates the seropositive cut-off for the neutralization titers (PsVNA50 of 40). Percent seropositivity (%S) for each group was calculated as number of seropositive samples divided by total number of samples ×100 in the group. The limit of detection for the neutralization assay is 1:20. Differences between the age groups within each SARS-CoV-2 strain were analyzed by lme4 and emmeans packages and Tukey-adjusted to reduce Type 1 error due to multiple testing in R and the significant *p*-values are shown. The tests were two-sided tests.

(Fig. S1b, c). However, for the youngest children, the WA-1 neutralizing titers were similar between the mRNA vaccination and the post-infection MIS-C and convalescent COVID-19 groups, which were significantly higher than acute COVID-19 cohort (Fig. S1a).

Neutralization titers against Omicron subvariants were rare for post-infection pediatric cohorts that were infected with the ancestral WA-1 or the Alpha strain (samples collected prior to March 2021) possibly due to the presence of low cross-reactive antibodies against circulating Omicron subvariants BA.2, BA.2.75, BA.3 and BA.4/BA.5 with seropositivity rate (PsVNA50 titer of ≥1:40) of 0–29% across all age groups and disease cohorts (Fig. 2a–c). Moreover, neutralizing antibody titers against the recently emerged highly transmissible BQ.1, BQ.1.1, and XBB.1 subvariants were further reduced with only 0–10% seropositivity across all post-infection cohorts (Fig. 2a–c).

In contrast to age-associated differences observed in neutralization titers against WA-1 that may be due to either vaccine dosage or the number of vaccine doses, the primary monovalent mRNA vaccination series in children induced similar cross-neutralizing antibody titers against the Omicron variants BA.2, BA.3 and BA.2.75 across all age groups (Fig. 2d). Surprisingly, younger children (<5 years old) demonstrated higher neutralizing antibody titers against newly emerged BQ.1 (GMT:62), BQ.1.1 (GMT:54), and XBB.1 (GMT:28) subvariants compared with older children (>5 years old) with GMT of only 11–21. When normalized to the neutralizing titers against the vaccine-homologous prototype WA-1 strain, the capacity of corresponding vaccination-induced antibodies to neutralize Omicron subvariants especially BQ.1, BQ.1.1 and XBB.1 was significantly higher for the younger children (<5 years) cohort compared with the older children (Fig. S2). The younger (<5 years) vaccinated children demonstrated seropositive titers (PsVNA50 titer of ≥1:40) against these highly transmissible BQ.1 (60%), BQ.1.1 (50%), and XBB.1 (20%) subvariants compared with only 5–10% of the vaccinated adolescents (Fig. 2d).

The findings of the anti-WA-1 neutralization titers in the post-infection and post-vaccination cohort were confirmed by the binding antibodies against receptor binding domain (RBD) of WA-1 spike protein using surface plasmon resonance (Fig. 3a and Fig. S3). As observed with neutralization titers, RBD-binding antibodies increased over time post-infection in younger children (<12 years) from acute COVID-19 disease to convalescent COVID-19 or children with MIS-C, as well as by age in the acute COVID-19 cohort. A good correlation was observed between WA-1 RBD binding antibodies and WA-1 neutralizing antibody titers for this pediatric post-infection and post-vaccination cohorts (Fig. 3b).

## Antigenic landscape following SARS-CoV-2 infection or vaccination in age-stratified children

To determine the relative antigenic relationship of the neutralizing antibodies against the different SARS-CoV-2 Omicron variants and the ancestral vaccine-homologous WA-1 across various age groups in different post-infection and post-vaccination cohorts we performed two-dimensional antigenic cartography (Fig. 4). In acute COVID-19 cohort, the Omicron variants are closest to WA-1 for the youngest age group

(<5 years) that start distancing apart by age from 5–11 years to adolescents. While for the convalescent COVID-19 and the MIS-C cohorts, the antigenic relationships are similar for the Omicron subvariants across age groups. In the vaccinated children, the youngest age group (<5 years) show close antigenic relationship between WA-1 and Omicron subvariants, while in the older children (>5 years), the Omicron subvariants have substantially diversified away from the vaccine-homologous WA-1, with the newly emerged highly transmissible BQ.1, BQ.1.1 and XBB.1 lineage most profoundly distinct among the Omicron subvariants. Bootstrap analysis was performed to assess the effect of uncertainty in the neutralization titers and variant reactivity (Fig. S4). Our analysis revealed the robustness of the SARS-CoV-2 antigen and serum relationships demonstrating the reliability of the antigenic cartography map despite the absence or explicit uncertainty bounds.

To quantify the magnitude of this antigenic relationship, we evaluated the antigenic landscape in these post-infection and post-vaccination children's cohorts (Fig. S5). An analysis of the three-dimensional antigenic landscape of three different age groups, corresponding with the neutralization titers presented in Fig. 2, revealed interesting findings (Fig. 5). As expected, in all age groups, mRNA vaccination induced higher cross-reactive neutralizing antibodies against variants than post-infection cohorts. In older children (5 years and older), the slope of the post-vaccination antigenic landscape and post-infection landscape was similar against the variants (Fig. 5b, c). Interestingly, the slope of the landscape for vaccination induced antibody response in the youngest children (<5 years) was shallower compared with much steeper slope observed for the vaccinated older children (5 years and older), suggesting a broader cross-reactive neutralizing antibodies following vaccination in this youngest age cohort (Fig. 5).

Within disease categories, both the magnitude of the neutralization titers and heterogeneity of antigenic diversity increases in age-dependent manner in the acute COVID-19 cohort (Fig. 6). The youngest age group (<5 years) shows low magnitude and the Omicron variants clustered closer to the ancestral WA-1, while acute COVID-19 adolescents showing greater heterogeneity in both titers and distance between WA1 and Omicron subvariants (Fig. S5). The convalescent COVID-19 and MIS-C samples display similar antigenic landscape across various age groups (Figs. 6, S5).

In the mRNA vaccinated children, the neutralizing antibodies induced following vaccination in the youngest age cohort, even though lower in magnitude against vaccine-homologous WA-1, are more cross-reactive against the circulating Omicron subvariants and are closer together on antigenic map (Figs. 6, S5). However, with increase in age, the magnitude of vaccination-induced anti-WA-1 titer also increased, but the antibody response in older children (>5 years) tends to be significantly less cross-reactive against Omicron subvariants especially the newly emerged XBB.1, BQ.1, and BQ.1.1 subvariants compared with younger children (<5 years), as demonstrated by the steep slope and spread of the antigenic landscape of the neutralization for the different variants, with a notable distance from WA-1 and other BA.2, BA.2.75, BA.3, and BA.4/BA.5 variants (Figs. 6, S5).

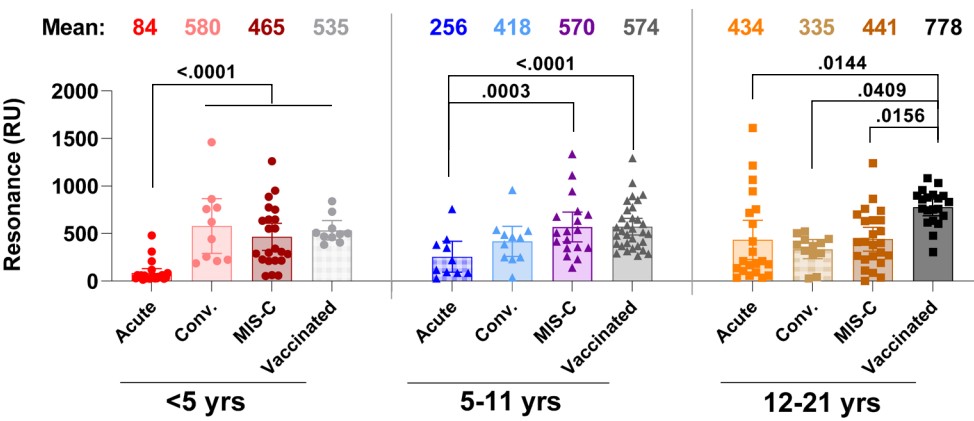

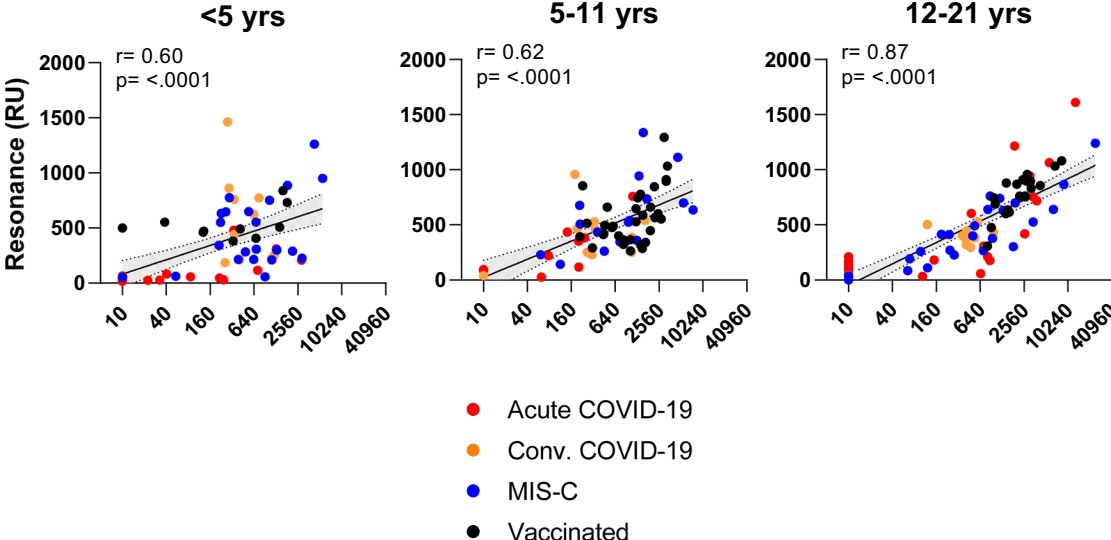

**Fig. 3 | Binding antibodies in serum/plasma of children with COVID-19 vs MIS-C vs vaccination to receptor binding domain of prototype vaccine-homologous SARS-CoV-2 WA-1 strain.** Total antibody binding (determined by maximum resonance units, Max RU) of 1:10 diluted serum or plasma to purified WA-1 spike RBD was measured by SPR. **a** Data is shown by age groups. Samples were divided by age categories: <5 years (*n* = 65; 23 acute, 10 convalescent, 22 MIS-C and 10 naïve vaccinated), 5–11 years (*n* = 71; 10 acute, 11 convalescent, 18 MIS-C and 32 naïve vaccinated) and 12–21 years old (*n* = 77; 21 acute, 12 convalescent, 24 MIS-C and 20 naïve vaccinated). The heights of the bars and the numbers over the bars indicate the mean antibody binding values and are color coded by group, and the whiskers indicate 95% confidence intervals. All SPR experiments were performed in duplicate and the researchers performing the assay were blinded to sample identity. The variations for duplicate runs of SPR were <5%. The data shown are average values of two experimental runs. The statistical significances between the variants were performed by lme4 and emmeans packages and Tukey-adjusted for multiple comparisons in R and the significant *p*-values are shown. The tests were two-sided tests. The differences were considered statistically significant with a 95% confidence interval when the *p* value was less than 0.05. The significant *p*-values are shown. **b** Relationship of RBD-binding antibodies and neutralizing antibodies against SARS-CoV-2 WA-1/2020 strain. Correlation of SARS-CoV-2 WA-1/2020 neutralizing titer versus WA-1 RBD binding antibodies measured by surface plasmon resonance for the three post-infection and the post-vaccinated children's cohorts. Correlations show Pearson correlation coefficient (r) and two-tailed *p* values for all samples. Black line in the scatter plots depict the linear fit of log2 transformed PsVNA50 values with shaded area showing 95% confidence interval.

## Discussion

There are different COVID-19 vaccine doses and regimens for various age groups of children and lower vaccination uptake in children especially in children who had prior SARS-CoV-2 infection. Therefore, it is critical to understand the capacity of post-infection and post-vaccination antibodies to neutralize currently circulating highly contagious SARS-CoV-2 BQ.1, BQ.1.1, and XBB.1 subvariants in an age-stratified manner and across disease spectrums in children, which can help refine the vaccine approach in this important but critically under-represented and under-studied population. In this pediatric study, we

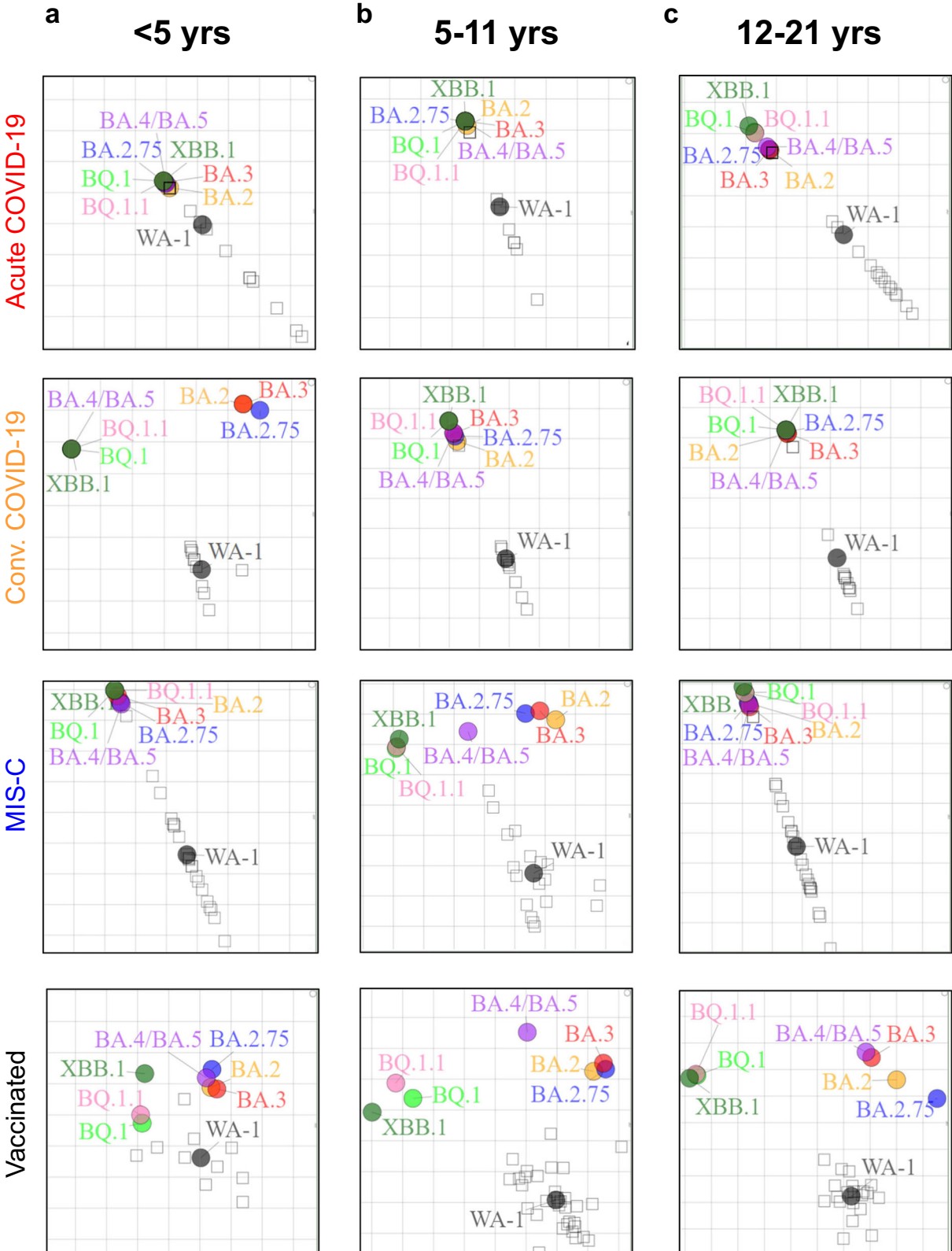

**Fig. 4 | Antigenic cartography of different age group children with acute COVID-19 vs convalescent COVID-19 vs MIS-C vs vaccination against SARS-CoV-2 WA-1 and Omicron subvariants.** Individual antigenic maps were generated for each infection or vaccination cohort in younger children (<5 years in 'a'), school-age children (5–11 years in 'b'), and adolescent (12–21 years in 'c'), with either acute COVID-19, convalescent COVID-19, MIS-C, or vaccination, against SARS-CoV-2 WA-1 the Omicron subvariants. Black diamonds correspond to each individual sera/plasma. One antigenic distance represented by each grid square corresponds to 2-fold dilution of neutralization assay.

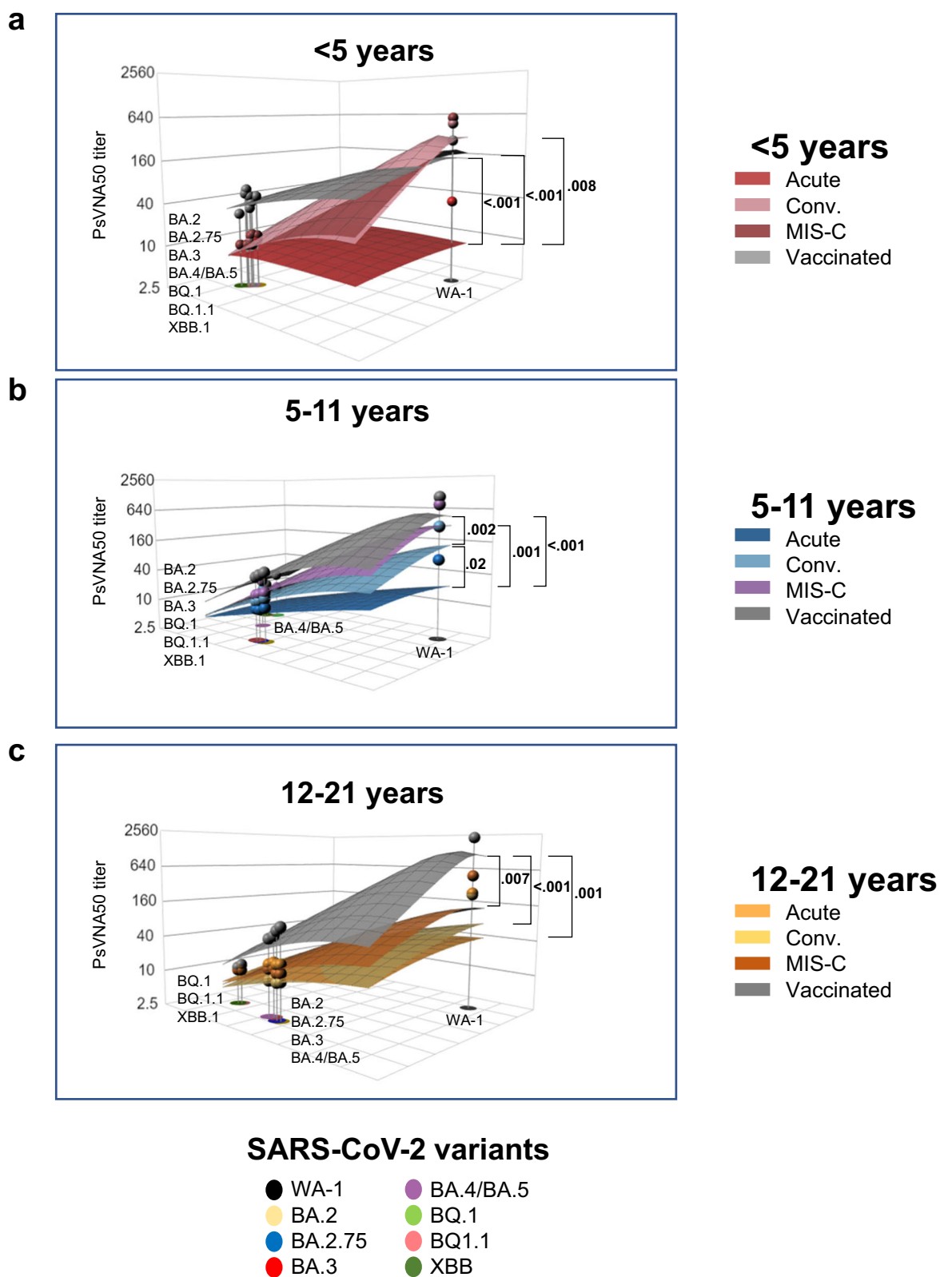

**SARS-CoV-2 variants**

- ● WA-1
- ● BA.2
- ● BA.2.75
- ● BA.3
- ● BA.4/BA.5
- ● BQ.1
- ● BQ1.1
- ● XBB

observed an age-associated effect in antibody response to SARS-CoV-2 in acute severe infection or vaccination, but not for convalescent mild COVID-19 or MIS-C. The acute COVID-19 cohort has less time to develop antibodies and antibody response may be suppressed due to acute viral disease. While the convalescent cohort who had mild disease and were never hospitalized with acute COVID-19, or the MIS-C who are estimated to be 3-6 weeks after their initial SARS-CoV-2 infection as it is a post-infectious complication, would have sufficient time to generate anti-SARS-CoV-2 immune response. We determined that ancestral WA-1 or Alpha infection-induced antibodies are not sufficient to neutralize highly contagious BQ.1, BQ.1.1, and XBB.1 subvariants across disease cohorts in any age group of children, and therefore vaccination would be key to prevent children against disease caused by the emerging SARS-CoV-2 variants. Surprisingly,

**Fig. 5 | Neutralizing antibody landscapes following infection or vaccination for different age groups of children.** The antigenic landscape was generated using the SARS-CoV-2 neutralization titers against WA-1 and Omicron BA.2, BA.3, BA.2.75, BA.4/BA.5, BQ.1, BQ1.1 and XBB.1 for the 213 children with either acute COVID-19, convalescent COVID-19, MIS-C or following vaccination, divided by age categories: (**a**) <5 years (*n* = 65; 23 acute, 10 convalescent, 22 MIS-C and 10 naïve vaccinated), (**b**) 5–11 years (*n* = 71; 10 acute, 11 convalescent, 18 MIS-C and 32 naïve vaccinated) and (**c**) 12–21 years old (*n* = 77; 21 acute, 12 convalescent, 24 MIS-C and 20 naïve vaccinated). The base *x* and *y* axis of each landscape map represents antigenic cartography between WA-1 and variants, with colored points representing the locations of each SARS-CoV-2 strain. The grid squares (1 antigenic unit) correspond to a 2-fold change in neutralization assay. The vertical *z*-axis corresponds to the neutralization titer on the $\log_2$ scale. The overall neutralizing antibody landscape for each pediatric group was constructed by fitting individual neutralizing antibody landscapes for each children's serum sample for that group against the respective SARS-CoV-2 strain. The landscapes are color coded according to patient categories (acute, convalescent, MIS-C, and vaccinated). The magnitude of neutralizing titers (PsVNA50 GMTs) for children either following vaccination or infection against each SARS-CoV-2 strain are shown by impulses connected for each SARS-CoV-2 variant on the *z*-axis in the landscape and the filled-circle symbols are color-coded by each patient category group. A Mann-Whitney test in R package was used to determine the significance of the difference between patient cohort for each age group, and the significant *p*-values are shown. The tests were two-sided tests.

mRNA vaccination of naïve children demonstrated an age-dependent or mRNA dose-dependent increase in neutralization titers against the vaccine-homologous WA-1 strain. However, this phenomenon was reversed against the newly emerged BQ.1, BQ.1.1, and XBB.1 subvariants, with the youngest children demonstrating more cross-neutralizing antibody titers than adolescents.

Following mRNA vaccination, we observed higher homologous WA-1 neutralization titers in older children compared with younger children. In contrast, the breadth of the neutralization response to SARS-CoV-2 variants relative to WA.1 reactivity, appears to be highest in the youngest age group (<5 years) compared with older children. Our study findings suggest the quality of antibody response induced following SARS-CoV-2 vaccination in young children may be less vulnerable to mutations in spike of the emerging SARS-CoV-2 variants in contrast to older children. One possible explanation for these qualitative antibody differences between age groups could be due to three doses of the 3 mcg mRNA vaccine in the primary vaccination schedule for these youngest children (<5 years) compared with two vaccine doses of higher dosage (10 or 30 mcg mRNA) for the older children (>5 years). Clinical studies in young children <5 years showed that three vaccinations with low mRNA dose (3 mcg mRNA) were required to generate equivalent neutralizing antibody response against the vaccine-matched ancestral WA-1 strain, as was induced by two doses (30 mcg mRNA) in older children[17]. It is possible that a third vaccine dose in children can improve the breadth of immune response against variants similar to the observations of broader cross-reactivity to Omicron variants after third versus second vaccine doses in adults[18]. In future studies, we plan to investigate the impact of the vaccine doses (three versus two) and hybrid immunity on the breadth of reactivity in an age stratified child cohort.

The other possible mechanism can be due to the antigenic sin hypothesis, whereby older children due to their prior exposure to seasonal human coronaviruses (hCoVs) have long-lived memory B cells specific for conserved sites on the coronavirus spike, especially in SARS-CoV-2 spike S2 domain[19–22]. Following vaccination, these memory B cells are recalled, thereby focusing the antibody response to these conserved sites in spike. Most of these cross-reactive antibodies to seasonal CoVs do not neutralize SARS-CoV-2, as was observed for children with hCoV infection[13]. Majority of the SARS-CoV-2 neutralizing antibody epitopes target the S1 domain of the SARS-CoV-2 spike that contains the receptor binding domain. However, since younger children (<5 years) have a shorter history of exposure to seasonal hCoVs, following vaccination they potentially could generate a more diverse antibody repertoire than immune-experienced older children, leading to higher cross-neutralization and potential efficacy against emerging SARS-CoV-2 highly contagious BQ.1, BQ.1.1 and XBB.1 subvariants. In support of this hypothesis, we observed that vaccination induced antibodies in younger children (<5 years) are more cross-reactive and effective in neutralizing Omicron BQ.1, BQ.1.1 and XBB.1 subvariants when normalized to the corresponding vaccine-homologous prototype WA-1 strain activity, compared with the older children (>5 years) (Fig. S2). Similar findings were observed of pandemic influenza vaccine studies in different age groups of children and adults who were given two doses of same vaccine dosage with or without MF59 adjuvant, where older children or adults contain pre-existing immunity against seasonal influenza strains. Following two doses of MF59-adjuvanted 7.5 mcg H1N1pdm09 vaccine, a higher diversity and affinity maturation of antibody repertoire was observed in younger children (12–35 months) against the HA1 domain of H1N1pdm09 compared with older children (3–8 years) or adults (18–60 years)[23]. This study suggested an impact of antigenic sin hypothesis of non-neutralizing antibody cross-reactivity to conserved epitopes in HA2 domain on the development of novel anti-H1N1pdm09 HA1 specific immune response following vaccination. In the current study, multiple attributes, including antigenic sin, vaccine dosage, number of vaccine doses, and the development stage of immune system, can all contribute to the age-associated differences observed following SARS-CoV-2 mRNA vaccination in these cohorts of children.

The presence of pre-existing immunity to seasonal CoVs, viral kinetics, as well as the developmental stage of the immune system in these various age children's cohorts may contribute to kinetics and magnitude of immune response following acute SARS-CoV-2 infection. This age-dependent phenomenon was observed at early time-point following infection in acute COVID-19 cohort, where older children developed faster and higher neutralizing antibody response than younger children against the infecting WA-1 strain. At later time-point post-exposure (convalescent COVID-19 and MIS-C), the kinetics of antibody response in the younger children reaches similar levels as the neutralization titers of older children. Moreover, the infection induced antibody response in these three age groups at 3 to 6-weeks post-infection (convalescent COVID-19 and MIS-C), comparable to the 4-week time-point post-vaccination, showed similar neutralization antibody response against the infecting ancestral WA-1 strain as well as Omicron subvariants. These differences observed between infection vs vaccination induced breadth of immune response to Omicron subvariants in various age groups of children, and mechanism underlying this antibody repertoire, remain an area of intense investigation.

The BNT162b2 vaccine and previous infection were found to confer considerable immunity against Omicron BA.1 infection and protection against hospitalization and death[10,11]. The rapid decline in protective neutralization titers against circulating BQ.1, BQ.1.1 and XBB.1 subvariants conferred by previous infection provides support for vaccination of children. As new SARS-CoV-2 variants emerge, continued determination of neutralizing antibodies will be needed in these children. Further studies are needed to characterize the neutralizing antibody response of children who received Omicron-based bivalent booster vaccinations. Unfortunately, the uptake of booster vaccination rates of children in US remains <8% (https://covid.cdc.gov/covid-data-tracker/#vaccination-demographics-trends).

Children remain at the epicenter of viral transmission, and therefore induction of broadly cross-neutralizing antibodies by current and new vaccination approaches that can reduce/block the transmission cycles are required to limit further evolution of new more transmissible and antibody-resistant SARS-CoV-2 variants. It is

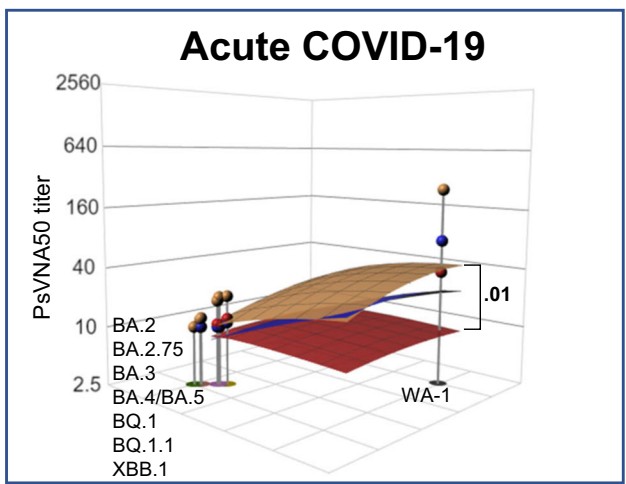

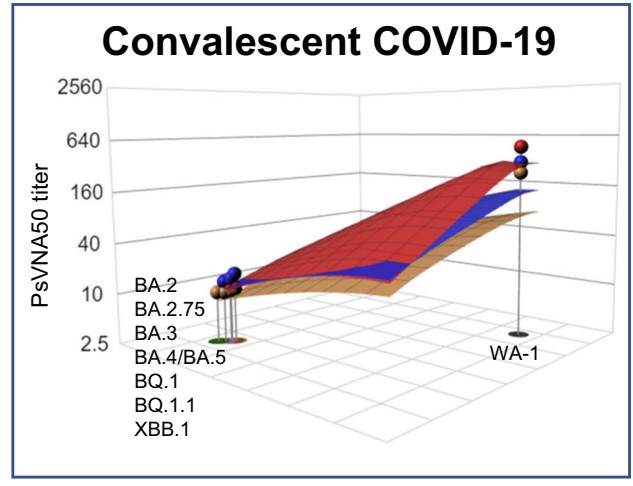

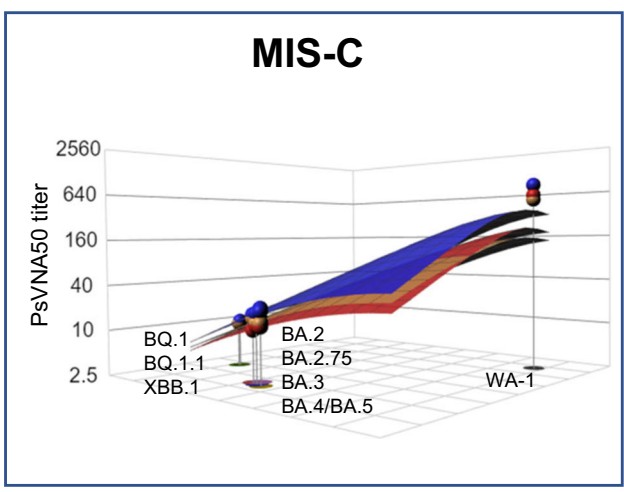

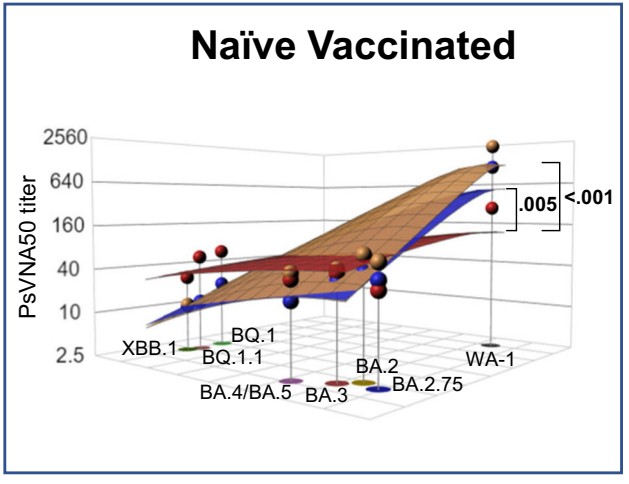

**Age Group**

- **<5** (dark red)
- **5-11** (blue)
- **12-21** (light orange)

**SARS-CoV-2 variants**

- WA-1
- BA.2
- BA.2.75
- BA.3
- BA.4/BA.5
- BQ.1
- BQ1.1
- XBB

**Fig. 6 | Comparison of age-stratified antigenic landscapes in different post-infection and post-vaccination children's cohorts.** The children's cohorts were divided into four categories: acute COVID-19, convalescent COVID-19, MIS-C, and naïve vaccinated. The antigenic landscape was generated using the SARS-CoV-2 neutralization titers against WA-1 and Omicron BA.2, BA.3, BA.2.75, BA.4/BA.5, BQ.1, BQ1.1 and XBB.1 for the 213 children with either acute COVID-19 ($n = 54$), convalescent COVID-19 ($n = 33$), MIS-C ($n = 64$) or following vaccination ($n = 62$) in different age groups. The landscapes are color coded according to age-stratified children. The $x$ and $y$ axis of each landscape map represent 2D antigenic cartography between WA-1 and variants, with colored points representing the locations of each SARS-CoV-2 strain. The grid squares (1 antigenic unit) represent antigenic distance that corresponds to a 2-fold change in neutralization titer. The vertical $z$-axis corresponds to the neutralization titer on the $\log_2$ scale. The overall neutralizing antibody landscape for each cohort was constructed by fitting individual neutralization landscapes for each pediatric serum sample for that age group against the respective SARS-CoV-2 strain. The magnitude of neutralization titers (PsVNA50 GMTs) for different age groups of against each SARS-CoV-2 variant are shown by impulses connected for each SARS-CoV-2 variant on the $z$-axis in the landscape, and the filled-circle symbols are color-coded by each age group. A Mann-Whitney test in R package was used to determine the significance of the difference between children age groups for each patient cohort, and the significant $p$-values are shown. The tests were two-sided tests.

---

plausible that a suboptimal neutralizing antibody response against circulating SARS-CoV-2 variants could potentially drive antigenic evolution through increased immune pressure in this population. Our data can help provide guidance on when children should be vaccinated and in designing policies for vaccination based on the immune status in settings where the risk of transmission is high, such as high-occupancy indoor setting in schools and daycare. The data suggests that an additional vaccine dose of the new bivalent or XBB-monovalent vaccine formulation would help broaden protection against newly emerging SARS-CoV-2 variants in children. Even children with prior SARS-CoV-2 infection will immensely benefit from hybrid immunity following vaccination to efficiently protect against circulating and emerging SARS-CoV-2 variants. Importantly, with over 3.6 million infants born in the U.S each year, the population of COVID-19 naïve individuals will continue to grow. Vaccination remains an important, effective strategy in protecting against severe COVID-19 and MIS-C. This study demonstrates that effective vaccine approaches are required to generate more broadly durable protective immunity

against COVID-19 or MIS-C caused by circulating Omicron BQ.1, BQ.1.1 and XBB.1 subvariants, which could influence the transmission and clinical outcome in the pediatric population.

## Methods

### Ethics statement

Collection of samples from vaccinated children received ethics approval from the MassGeneral Brigham Institutional Review Board (IRB) (MGB #2020P000955). Collection of multicenter samples from unvaccinated children received ethics approval from the single IRB at Boston Children's Hospital (BCH # P00033157). Informed consent, and assent when appropriate, was obtained from all participants and/or parents/legal guardians. All methods and procedures were approved by the IRB and carried out in accordance with the IRB's guidelines and regulations. The study at CBER, FDA, was conducted with de-identified samples. This study was approved by the Food and Drug Administration's Research Involving Human Subjects Committee (RIHSC #2020-04-02). This study complied with all relevant ethical regulations for work with human participants, and informed consent was obtained. All assays performed fell within the permissible usages in the original informed consent.

### Study design

The objective of this study was to investigate the neutralizing capacity of serum or plasma from children of age-stratified groups with acute, convalescent COVID-19 or MIS-C compared with vaccinated naïve children against circulating SARS-CoV-2 variants.

The 151 U.S. pediatric patients were enrolled across multiple hospitals in the Overcoming COVID-19 Network comprised 3 independent cohorts: 54 hospitalized with acute COVID-19, 64 hospitalized with MIS-C, and 33 non-hospitalized children with mild SARS-CoV-2 infection with convalescent samples (≥30 days post-acute) enrolled as an at-risk for MIS-C control group, all divided in different age groups (Fig. 1b). MIS-C and acute COVID-19 were defined using U.S. Centers for Disease Control and Prevention (CDC) case definitions (https://wwwn.cdc.gov/nndss/conditions/coronavirus-disease-2019-covid-19/case-definition/2020/ approved April 5th, 2020 and https://www.cdc.gov/mis-c/hcp/ published 5/14/2021). All acute and convalescent COVID-19 pediatric patients had SARS-CoV-2 detected by reverse transcriptase quantitative PCR (RT-qPCR) and MIS-C patients had positive SARS-CoV-2 antibody and/or RT-qPCR tests. Samples from the post-infection pediatric cohorts (acute COVID-19 vs convalescent COVID-19 vs MIS-C) were collected from a prospectively enrolling multicenter study (Overcoming COVID-19). Children's acute COVID-19 samples (serum or plasma) were collected as early as possible during hospitalization and/or study enrollment. The demographic for the pediatric cohort is shown in Table S2. The clinical information for the post-infection pediatric cohorts were described before[13].

Fresh blood was collected into sodium heparin, EDTA, or no additive vacutainers and centrifuged at $1300 \times g$ (RCF) for 10 min at room temperature. Plasma or serum was aliquoted and frozen at −80 °C. If a fresh blood sample was not obtained upon enrollment, residual specimens from clinical testing (lithium heparin plasma or serum) were retrieved. Samples were heat-treated at 56 °C for 1 h and refrozen at −80 °C prior to assay.

Serum samples were also obtained from 62 naïve (SARS-CoV-2 negative) children who were vaccinated with two or three doses of Pfizer's BNT16b2 mRNA monovalent vaccine encoding WA-1 spike. The youngest children (<5 years) received three doses of 3 mcg mRNA vaccine, while children aged 5–11 years got 2 doses of 10 mcg mRNA vaccine, and adolescents (12–21 years) received 2 doses of 30 mcg mRNA vaccine at three weeks apart. Samples were collected at 28 days after completion the primary vaccination series from these children.

All children irrespective of sex, race, ethnicity, or previous health status that were exposed to SARS-CoV-2 or COVID-19 vaccination were eligible for the study. Sex, race, ethnicity, or previous health status was not considered in the study design and study findings apply to children of both sexes or any race or ethnicity. The sex, race, ethnicity, and previous health status was determined based on self-reporting. The sex distribution, race, ethnicity, and previous health status in each cohort is shown in Table S2. No sex-based or race-based analyses were performed in this study.

### Lentivirus pseudovirion neutralization assay (PsVNA)

Samples were evaluated in a qualified SARS-CoV-2 pseudovirion neutralization assay (PsVNA) using SARS-CoV-2 WA1/2020 strain and circulating Omicron subvariants: BA.2, BA.2.75, BA.3, BA.4/BA.5, BQ.1, BQ.1.1 and recombinant XBB.1. The mutations in spike protein of these Omicron subvariants are shown in Table S1. SARS-CoV-2 neutralizing activity measured by PsVNA correlates with PRNT (plaque reduction neutralization test with authentic SARS-CoV-2 virus) in previous studies[14,15,24].

Neutralization assays were performed as previously described[13–15,25,26]. Briefly, 50 μL of SARS-CoV-2 Spike pseudovirions (counting ~200,000 relative light units) were pre-incubated with an equal volume of medium containing serial dilutions (starting at 1:10) of all samples at room temperature for 1 h. Then, 50 μL of virus-antibody mixtures were added to 293T-ACE2-TMPRSS2 cells[24] [$10^4$ cells/50 μL] in a 96-well plate. The input virus with all SARS-CoV-2 strains was the same ($2 \times 10^5$ relative light units/50 μL/well). After a 3 h incubation, fresh medium was added to the wells. Cells were lysed 24 h later, and luciferase activity was measured using One-Glo luciferase assay system (Promega). The assay of each sample was performed in duplicate, and the 50% neutralization titer was calculated using Prism 9 (GraphPad Prism Software). The limit of detection for the neutralization assay is 1:20. Two independent biological replicate experiments were performed for each sample and variation in PsVNA50 titers was <10% between replicates.

### Antibody binding kinetics to SARS-CoV-2 RBD by surface plasmon resonance (SPR)

Purified recombinant SARS-CoV-2 spike receptor binding domain (RBD) of WA-1 strain expressed in HEK293 Cells was purchased from Sino Biologicals (wtRBD; 40592-V08H). Steady-state equilibrium binding in serum/plasma samples was monitored at 25 °C using a ProteOn surface plasmon resonance (BioRad). The purified recombinant SARS-CoV-2 RBD proteins were captured to a Ni-NTA sensor chip with 500 resonance units (RU) in the test flow channels[14,20,21,27].

Serial dilutions (10-, 30- and 90-fold) of freshly prepared sample diluted in BSA-PBST buffer (PBS pH 7.4 buffer with Tween-20 and Bovine Serum Albumin) were injected at a flow rate of 50 μL/min (120 s contact duration) for association, and disassociation was performed over a 600-second interval. Responses from the protein surface were corrected for the response from a mock surface and for responses from a buffer-only injection. Total antibody binding was calculated with BioRad ProteOn manager software (version 3.1). All SPR experiments were performed twice, and the researchers performing the assay were blinded to sample identity. The variations for duplicate runs of SPR were <5%. The maximum resonance units (Max RU) shown in the figures were the calculated RU signal for the 10-fold dilution sample.

### Statistical analysis

Descriptive statistics were performed to determine the geometric mean titer values and were calculated using GraphPad Prism. All experimental data to compare differences among groups were analyzed using lme4 and emmeans packages in R (R version 4.1.2).

Since sex can be biologically plausible confounders, data were analyzed for statistical significance between groups to control for sex as covariate (predictor variables) using a multivariate linear regression

model. To ensure robustness of the results, absolute measurements were log2-transformed before performing the analysis. For comparisons between the age and treatment (vaccine/disease) cohorts (factor variable), pairwise comparisons were extracted using 'emmeans' and Tukey-adjusted p values were used for denoting significance to reduce Type 1 error due to multiple testing. The tests were two-sided tests. The differences were considered statistically significant with a 95% confidence interval when the p value was less than 0.05.

Samples were allocated randomly to each test group and tested blindly (researcher was blinded to sample identity) to minimize selection bias or detection bias. There were no exclusion criteria. All samples and data were used for analysis and presented in the study.

### Antigenic landscape and cartography
Antigenic cartography analysis was performed using the Racmacs package in R software. Multi-dimensionality antigenic landscape maps were developed using the Rossler method, which measures the antigenic distance as well as the magnitude of neutralization titers against WA-1 and variants, was constructed as described before[28–30].

To evaluate the robustness of our findings and account for potential variation in antigenic cartography, bootstrap analysis was conducted with 1000 bootstraps repeats and 100 optimizations per repeat, as described before[30], using the R package (Racmacs). The bootstrap analysis incorporated noise by adding normally distributed values to both the titers and antigen reactivity. The standard deviation of the noise added to the neutralization titers was 0.7, while the standard deviation for the noise added to antigen reactivity was 0.7.

In order to assess the significance of the landscape between different cohorts in the antigenic map, we performed a statistical analysis using Mann-Whitney test in R. The purpose of the test was to determine the relationship between age groups and patient categories in the antigenic landscape.

### Reporting summary
Further information on research design is available in the Nature Portfolio Reporting Summary linked to this article.

## Data availability
All data are shown in the manuscript figures and supplementary information. The complete dataset for this study is provided in the Source Data file. Source data are provided with this paper.

## Code availability
Antibody titers were calculated using Prism 9.3.1 (GraphPad Software). Statistical analyses were performed using R statistical software (version 4.1.2). No new code was generated.

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

## Acknowledgements

We thank Hana Golding and Keith Peden for review of the manuscript. We thank Carol Weiss for providing plasmid clones expressing SARS-CoV-2 variants. The research work described in this manuscript was supported by FDA Perinatal Health Center of Excellence (PHCE) grant #GCBER005 and #GCBER008 to S.K., and the U.S. Centers for Disease Control and Prevention (contracts #75D30120C07725, 75D30121C10297 and 75D30122C13330) to A.G.R.; Patient clinical data and specimens also collected at Boston Children's Hospital for the Taking on COVID-19 Together (TOCT) study supported in part by the Boston Children's Hospital Emerging Pathogens and Epidemic Response Cluster of Clinical Research Excellence and the Institutional Centers for Clinical and Translational Research to A.G.R. The National Heart Lung and Blood Institute (5K08HL143183 to L.Y.) at Mass General for Children and Boston Children's Hospital. The funders had no role in study design, data collection and analysis, decision to publish, or preparation of the manuscript. The content of this publication does not necessarily reflect the views or policies of the Department of Health and Human Services or the Centers for Disease Control and Prevention, nor does mention of trade names, commercial products, or organizations imply endorsement by the U.S. Government.

## Author contributions

Designed research: S.K., A.R., L.Y. Clinical specimens and clinical data: T.N., A.R., L.Y. Performed assays: L.B., G.G., and S.K. Statistical analyses: G.G. and S.S. Contributed to Writing: S.K. Review and Editing of the manuscript: S.K., A.R., T.N., L.Y.

## Competing interests

The authors declare no competing interests.

## Additional information

## Overcoming COVID−19 Investigators

**Adrienne G. Randolph** [3,4], **Tanya Novak** [3,4] **& Takuma  Kobayashi** [4]

