## [Peer Review File · Nature Communications]

REVIEWER COMMENTS

Reviewer #1 (Remarks to the Author):

This paper provides an informative comparative analysis of patterns of neutralization of post-infection and post-vaccination sera from children of different age groups against WA.1 and more recently circulating omicron variants. However, some differences attributed to age-related differences may be confounded by other factors.

When comparing between the different categories of infection sera one important factor that could be better described is how soon after presentation or symptom onset the different samples were taken and how this affected the magnitude and pattern of titers. In the main text it is stated that:

"Acute COVID-19 samples (serum or plasma) were collected as early as possible during hospitalization and/or study enrollment and within 2 weeks of presentation."

This is important because a lot of the variation seen in the acute infection groups could be dependent upon exactly when samples were taken since antibody response dynamics mean that magnitude of response can change greatly depending on if the sample is taken 5 days or 2 weeks after symptom onset. For example, later where it is stated:

"For the post-infection cohort, among pediatric acute COVID-19, a steady increase of neutralizing antibodies against the ancestral WA-1 infecting strain was observed as age increased."

Is this simply because acute samples were taken slightly later in the course of infection for the eldest age group and can this be controlled for if so? Also, it is unclear how long after initial presentation MIS-C samples were taken.

Separately is difficult to draw conclusions from several of the antigenic maps produced in figure S3 since no uncertainty bounds are shown in any figures for the relative positions of each of the variants according to each of the sera. In some cases there is a completely linear pattern of antigen and serum points that seems to be related to cases where there are a large number of non-detectable titers against several of the omicron variants and therefore presumably great uncertainty in determining antigenic differences between them according to the sera in those maps. In this regard the more informative antigenic maps are those where sera have a larger degree of detectable titers, in particular the

vaccinated serum groups, where the condensed nature of the antigenic map in the <5 yrs group is reflective of the very interesting observation that this group tends to have increased breadth of cross-reactivity to the new omicron variants after accounting for lower overall titer magnitudes. The same comments apply to the extension of the maps as landscapes in Figure 4, where in particular the landscape that spreads into 2-dimensions is likely not an informative representation in the cases where the variants have a completely 1-dimensional linear arrangement.

Lastly, in the discussion, when drawing conclusions about the interesting differences in reactivity patterns between the different post-vaccination age-groups, it would be useful to more explicitly distinguish between differences in response magnitude (where homologous WA.1 titers are found to be higher in older children who receive a larger dose) and response breadth (reactivity to other variants relative to WA.1 reactivity, which appears to be highest in the youngest age group).

With regard to the increased breadth of response in the youngest cohort, a lot of discussion is dedicated to an original antigenic sin hypothesis, however additional references could be made to findings in adults that increased response breadth is also similarly apparent after 3x vs 2x vaccine doses. It would be interesting to compare the breadth of response seen in the youngest cohort after receiving only 2 doses of the vaccine regimen, which would help distinguish between the two hypotheses of number of doses vs effect of previous exposures. Similarly, it would be interesting to compare how the breadth of reactivity in the 3x vaccinated children compares to the breadth seen in adults after 3 doses, although I appreciate this may be outside the scope of this particular study.

Overall, given findings in adults that vaccine dose is related to increasing response magnitude and that a 3rd dose may be related to increasing response breadth, it seems likely that these factors alone may influence the differences seen in the different vaccination age groups. The authors acknowledge this to some degree but it seems like an important observation to make since if this is indeed the main driver of the age-group differences seen does this imply that utilising larger doses for younger age groups and an additional dose for older age-groups would help broaden protection against newly emerging variants?

Minor comments:

Line 111:

"Acute children <5 years old also required significantly less respiratory support than adolescents with the same acute illness ($p=0.02$). There were no differences in disease severity based on ICU admission, respiratory support, or mechanical ventilation among different MIS-C age groups."

More accurate to say "no significant differences found", also it is not clear that any attempt was made to account for multiple testing using Pearson's chi-squared test.

This sentence on line 147 is very difficult to understand:

"The seropositivity rate against WA-1 (PsVNA50 titer of >1:40) predicted to provide protection against severe COVID-19 16 increased from 30% for acute COVID-19 patients to 95-100% for the convalescent COVID-19 for the MIS-C samples in the youngest age cohort (Figure S1a)."

Line 44:

"Monovalent mRNA vaccination induced more cross-neutralizing antibodies against SARS-CoV-2 variants especially BQ.1, BQ.1.1 and XBB.1 in young children <5 years compared with ≥5 years old children."
More cross-neutralizing compared to what?

Line 43:

"In contrast, BQ.1, BQ.1.1 and XBB.1 subvariants are more resistant".

Were not so well neutralized?

Line 55:

"widespread COVID-19 around the globe"

circulation?

Line 265:

"Children remain at the epicenter of viral transmission, therefore, induction of broadly cross-neutralizing antibodies by current and new vaccination approaches can reduce/block the transmission cycles to limit further evolution of new more transmissible and antibody-resistant SARS-CoV-2 variants."

It is also possible that this could drive further/quicker antigenic evolution through increased immune pressure in this population.

Reviewer #2 (Remarks to the Author):

This study compares two groups of children; i) age-stratified children with acute, severe COVID-19 (n=151) and ii) age-stratified vaccinated (BNT162b2) children (n=62). Neutralising antibody titres were examined against Omicron variants circulating in recent months (BA.4/BA.5, BA.2.75, BQ.1, BQ.1.1 and XBB.1).

Findings:

Vaccinated adolescents had higher titres against ancestral strain than <12 years.

Infected children had low titres against BA.2., BA.3, BA.4/BA.5, and BA.2.75, and minimal activity against BQ.1, BQ.1.1 and XBB.1.

Vaccination induced higher titres against variants than infected children, with measurable, but low activity against BQ.1, BQ.1.1 and XBB.1.

As the post-infection samples were collected between April 2020 and March 2021, when the prevalent circulating lineages in the US were the ancestral strain followed by the Alpha variant, it is unclear what value can be discerned from studying cross-neutralisation of Omicron-derived lineages in this population (perhaps this could be clarified).

The sera collected from vaccinated children were all collected approximately 20 days post vaccination, and comparisons between the three age groups suggest an age-related increase in antibody titres. The comparison between cohorts (acute COVID-19, MISC and convalescent) is difficult as acute samples were collected within 2 weeks of presentation while convalescent sera were approximately 60-70 days post positive test. Hence, the immune response in the acute cohorts will have had less time to develop and may be suppressed due to acute viral disease. Age-stratified within cohort comparisons would be valid, and an age-related increase in titres was observed in the acute cohort.

In the convalescent cohort, titres were similar across all ages.

Antigenic cartography demonstrated that, irrespective of the cohort, the Omicron lineages and the ancestral variant fall into two distinct neutralisation groups. In the acute cohort, the data suggest low magnitude and heterogeneity of the neutralising response in the youngest group, while convalescent samples showed similar magnitude and heterogeneity across age groups. The analysis of "antigenic landscape" was not quantified, hence there is no statistical confidence in the conclusions. Could the antigenic landscape data be compared statistically?

While the analyses of the neutralising activity and antigenic cartography are both valid and interesting, the rationale for comparing post-exposure sera from early in the pandemic, with post-vaccination sera to look at cross-neutralisation of late pandemic variants is not clear. The main conclusion is that for these cohorts, vaccination induced a more broad neutralising response than infection, further

comparisons between the cohorts are likely confounded by the mis-matched dates of sampling post-exposure.

Reviewer #3 (Remarks to the Author):

This manuscript described the neutralising antibody responses against the current circulating omicron subvariants in children and adolescents previously infected with the ancestral Wuhan or Alpha strain. The authors found that the age of children/adolescent may influence the magnitude and cross-reactivity of neutralising antibody responses following vaccination or infection. The strength of the paper is the unique samples cohort that comprises of children/adolescents samples with a range of clinical diseases and ages. I have several queries and comments on the paper:

1. What is the context/relevance of comparing acute vs MISC or convalescent or vaccinated? It is understood that acute samples would generally be lower than convalescent. In addition, Line 142-143: please remove “demonstrated an increase in WA-1 neutralizing antibodies in younger children”. They are not directly comparable to acute patients since they are different individuals. It might indicate that children under 5 takes longer to mount full antibody response. Furthermore, there was no increase in antibody levels between samples from acute and convalescent adolescents, suggesting that samples collected during acute phase may be almost convalescent samples?

2. How do you define MISC when acute patients also requires ICU admission and respiratory support? MISC usually occurs 2 weeks after covid-19- how does that influence the timing of blood sampling- please clarify.

3. It is a little misleading to say vaccinated children under 5 demonstrate more cross-reactivity than older children/adolescents since children under 5 years old received 3 doses while children aged ≥ 5 years old as well as adolescents up to 21 years old received 2 doses. In adults, 3 doses are needed to generate neutralising antibodies against omicron variants. Therefore, the neutralising antibody responses may not be directly comparable in terms of magnitude and cross-reactivity between the different age group who received different doses of vaccine. This may influence the results for Figure 4/antigenic landscape if they are not comparable

4. Discussion of antigenic sin was good but other parts needs improvement. E.g. Line 268-270 is very vague. Please provide details how your findings can provide guidance which leads to my next point of study limitations- please include. The conclusion do not align with the data. E.g. what effective vaccine

approaches can be derived from this study? This study applies to alpha- and/or wuhan-related infection. Hybrid immunity and immunity from omicron-infected children may be different.

5. Line 147-150: “The seropositivity rate against WA-1 (PsVNA50 titer of >1:40) predicted to provide protection against severe COVID-19 increased from 30% for acute COVID-19 patients to 95-100% for the convalescent COVID-19 for the MIS-C samples in the youngest age cohort (Figure S1a)”. I would suggest to remove or reword these sentences. The reference correlate is for vaccination and may not apply to all variants.

6. Neutralising antibodies against the omicron subvariants in Figure 2 is hard to visualise/interpret with text. Assuming GMT10 is the minimum detection limit and anything under GMT40 is seronegative. The text should mention seronegativity/seropositivity rather than GMT10-12 or 10-22, which is negative. Please consider revising the text for results (line 171-line 187). I would also suggest using seropositivity rather than protective titers.

7. Please consider rearranging results presentation- there are reference to figure 2 and then to figure 3 and then back to figure 2. Perhaps the RBD binding data can be presented first to improve readability or move to supplementary.

Minor comments

Line 43-44: please consider rewording.

Line 71 and line 263: would that be a reference

Line 72: please provide some figures for context.

Line 84: edits required

Results: Please provide median age for each age group, and actual number and percentage, e.g. X/X (X%) for line 108-113.

Line 84-88: rewording required.

Line 223-228: long sentence

Line 231: please indicate wuhan or alpha infection-induced antibodies..

Line 323 and 325: which recombinant RBD protein was used- sinobiological or in-house made.

REVIEWERS' COMMENTS:

Reviewer #1 (Remarks to the Author):

This paper provides an informative comparative analysis of patterns of neutralization of post-infection and post-vaccination sera from children of different age groups against WA.1 and more recently circulating omicron variants. However, some differences attributed to age-related differences may be confounded by other factors.

When comparing between the different categories of infection sera one important factor that could be better described is how soon after presentation or symptom onset the different samples were taken and how this affected the magnitude and pattern of titers. In the main text it is stated that:

"Acute COVID-19 samples (serum or plasma) were collected as early as possible during hospitalization and/or study enrollment and within 2 weeks of presentation."

This is important because a lot of the variation seen in the acute infection groups could be dependent upon exactly when samples were taken since antibody response dynamics mean that magnitude of response can change greatly depending on if the sample is taken 5 days or 2 weeks after symptom onset. For example, later where it is stated:

"For the post-infection cohort, among pediatric acute COVID-19, a steady increase of neutralizing antibodies against the ancestral WA-1 infecting strain was observed as age increased."

Is this simply because acute samples were taken slightly later in the course of infection for the eldest age group and can this be controlled for if so? Also, it is unclear how long after initial presentation MIS-C samples were taken.

Response: We thank the reviewer for the positive response.

We have revised the Supplementary Table S2 and added information on the timing of sample collection for various cohorts. It is clear in Supplementary Table S2, that for acute COVID-19 patients, the median days the sample were taken from hospital admission is 1 day higher in the older group: 1 day (1, 2) for <5 years; 1 day (1, 2.5) for 5 to <12 years; 2 days (1, 4) for 12-21 years). However, the majority of samples for all groups were taken close to admission on average.

The MIS-C samples were taken slightly longer after admission than acute COVID, but it is a post-infectious complication that occurs approximately 3-6 weeks post-SARS-CoV-2 infection, and we do not know exactly when they were originally infected with SARS-CoV-2. The MIS-C samples were collected within same timeframe for different age groups from hospital admission: 3 days (1.3, 6) for <5 years; 2.5 days (1, 5) for 5 to <12 years; 4 days (2, 14.3) for 12-21 years).

We do not believe these differences in timing of sample collection explain the age-related differences in antibody response.

We have added the following information to reflect the timing of the sample in the revised manuscript.

Lines 114-124: There was no statistical difference in the timing of pediatric sample collection relative to hospital admission or PCR/antibody positive test between the various age groups in any of the disease cohorts (Table S2).

MIS-C is a hyperinflammatory syndrome that occurs approximately 3-6 weeks post-SARS-CoV-2 infection in children who were asymptomatic or had mild illness upon initial infection. As a control group for MIS-C, we collected convalescent COVID-19 samples from outpatient children and adolescents with asymptomatic or mild illness (never hospitalized) approximately 3-6 weeks after their positive SARS-CoV-2 PCR test. Because MIS-C is rare (~2-3 per 10,000 SARS-CoV-2 infections in US), and it is not possible to identify patients in advance, therefore, it is not feasible to get baseline (during initial infection) samples in most MIS-C patients.

Separately is difficult to draw conclusions from several of the antigenic maps produced in figure S3 since no uncertainty bounds are shown in any figures for the relative positions of each of the variants according to each of the sera.

In some cases there is a completely linear pattern of antigen and serum points that seems to be related to cases where there are a large number of non-detectable titers against several of the omicron variants and therefore presumably great uncertainty in determining antigenic differences between them according to the sera in those maps. In this regard the more informative antigenic maps are those where sera have a larger degree of detectable titers, in particular the vaccinated serum groups, where the condensed nature of the antigenic map in the <5 yrs group is reflective of the very interesting observation that this group tends to have increased breadth of cross-reactivity to the new omicron variants after accounting for lower overall titer magnitudes.

Response: In the context of antigenic maps, the distance is calculated using serological data obtained from neutralization assay, which measure the level of antibody cross-reactivity between pairs of strains. The strains that are more similar in terms of their antigenic properties are plotted closer together than strains that are more different.

We have addressed the concern of uncertainty bounds by applying “smooth” bootstrap approach to assess the effect of uncertainty in the neutralization titers and variant reactivity. The data is shown in new supplementary figure S4 in the revised manuscript. To assess the impact of uncertainty in titers and variant reactivity on the antigenic cartography map, a "smooth" bootstrap approach was performed using 1000 bootstrap repeats, with 100 optimizations per repeat. The bootstrap analysis incorporated noise by adding normally distributed values to both the titers and antigen reactivity. The standard deviation of the noise added to the neutralization titers was 0.7, while the standard deviation for the noise added to antigen reactivity was 0.7. Our analysis reveals that the antigen and serum positions show robustness even when considering uncertainty. This finding

supports the reliability of the antigenic map despite the absence or explicit uncertainty bounds.

Results Lines 225-228: Bootstrap analysis was performed to assess the effect of uncertainty in the neutralization titers and variant reactivity (Figure S4). Our analysis revealed the robustness of the SARS-CoV-2 antigen and serum relationships demonstrating the reliability of the antigenic cartography map despite the absence or explicit uncertainty bounds.

Methods Lines 445-450: To evaluate the robustness of our findings and account for potential variation in antigenic cartography, bootstrap analysis was conducted with 1000 bootstraps repeats and 100 optimizations per repeat, as described before³⁰, using the R package (Racmacs). The bootstrap analysis incorporated noise by adding normally distributed values to both the titers and antigen reactivity. The standard deviation of the noise added to the neutralization titers was 0.7, while the standard deviation for the noise added to antigen reactivity was 0.7.

The same comments apply to the extension of the maps as landscapes in Figure 4, where in particular the landscape that spreads into 2-dimensions is likely not an informative representation in the cases where the variants have a completely 1-dimensional linear arrangement.

Response: A landscape representation is intended to present a visual summary which combines both the antigenic relationship and the magnitude of the mean titers for the samples against the SARS-CoV-2 variants. To represent data more informatively, additional analyses were performed to compare the landscape of different disease cohort within each age group (new figure 5), as well as of each pediatric age group in different disease cohort (new figure 6). We have discussed the results further in revised manuscript.

Results Lines 230-257: To quantify the magnitude of this antigenic relationship, we evaluated the antigenic landscape in these post-infection and post-vaccination children's cohorts (Figure S5). An analysis of the three-dimensional antigenic landscape of three different age groups revealed interesting findings (Figure 5). As expected, in all age groups, mRNA vaccination induced higher cross-reactive neutralizing antibodies against variants than post-infection cohorts. In older children (5 years and older), the slope of the post-vaccination antigenic landscape and post-infection landscape was similar against the variants (Figure 5b-c). Interestingly, the slope of the landscape for vaccination induced antibody response in the youngest children (<5 years) was shallower compared with much steeper slope observed for the vaccinated older children (5 years and older),

suggesting a broader cross-reactive neutralizing antibodies following vaccination in this youngest age cohort (Figure 5).

Within disease categories, both the magnitude of the neutralization titers and heterogeneity of antigenic diversity increases in age-dependent manner in the acute COVID-19 cohort (Figure 6). The youngest age group (<5 years) shows low magnitude and the Omicron variants clustered closer to the ancestral WA-1, while acute COVID-19 adolescents showing greater heterogeneity in both titers and distance between WA1 and Omicron subvariants (Figure S5). The convalescent COVID-19 and MIS-C samples display similar antigenic landscape across various age groups (Figure 6 and S5).

In the mRNA vaccinated children, the neutralizing antibodies induced following vaccination in the youngest age cohort even though lower in magnitude against vaccine-homologous WA-1, are more cross-reactive against the circulating Omicron subvariants and are closer together on antigenic map (Figure 6 and Figure S5). However, with increase in age, the magnitude of vaccination-induced anti-WA-1 titer also increased, but the antibody response in older children (>5 years) tends to be significantly less cross-reactive against Omicron subvariants especially the newly emerged XBB.1, BQ.1, and BQ.1.1 subvariants compared with younger children (<5 years), as demonstrated by the steep slope and spread of the antigenic landscape of the neutralization for the different variants, with a notable distance from WA-1 and other BA.2, BA.2.75, BA.3, and BA.4/BA.5 variants (Figure 6 and Figure S5).

Lastly, in the discussion, when drawing conclusions about the interesting differences in reactivity patterns between the different post-vaccination age-groups, it would be useful to more explicitly distinguish between differences in response magnitude (where homologous WA.1 titers are found to be higher in older children who receive a larger dose) and response breadth (reactivity to other variants relative to WA.1 reactivity, which appears to be highest in the youngest age group).

Response: Thanks for the suggestion. We have added a new supplementary figure S2 that shows the neutralization data normalized to the WA-1 titers to clarify that youngest children demonstrate broader cross-reactive SARS-CoV-2 neutralization titers. We have added information in the results section.

Lines 196-200: When normalized to the neutralizing titers against the vaccine-homologous prototype WA-1 strain, the capacity of corresponding vaccination-induced antibodies to neutralize Omicron subvariants especially BQ.1, BQ.1.1 and XBB.1 was significantly higher for the younger children (<5 years) cohort compared with the older children (Figure S2).

This was also observed in comparative landscape analysis as shown in new figure 6.

Results Lines 241-257:

Within disease categories, both the magnitude of the neutralization titers and heterogeneity of antigenic diversity increases in age-dependent manner in the acute COVID-19 cohort (Figure 6). The youngest age group (<5 years) shows low magnitude and the Omicron variants clustered closer to the ancestral WA-1, while acute COVID-19 adolescents showing greater heterogeneity in both titers and distance between WA1 and Omicron subvariants (Figure S5). The convalescent COVID-19 and MIS-C samples display similar antigenic landscape across various age groups (Figure 6 and S5).

In the mRNA vaccinated children, the neutralizing antibodies induced following vaccination in the youngest age cohort even though lower in magnitude against vaccine-homologous WA-1, are more cross-reactive against the circulating Omicron subvariants and are closer together on antigenic map (Figure 6 and Figure S5). However, with increase in age, the magnitude of vaccination-induced anti-WA-1 titer also increased, but the antibody response in older children (>5 years) tends to be significantly less cross-reactive against Omicron subvariants especially the newly emerged XBB.1, BQ.1, and BQ.1.1 subvariants compared with younger children (<5 years), as demonstrated by the steep slope and spread of the antigenic landscape of the neutralization for the different variants, with a notable distance from WA-1 and other BA.2, BA.2.75, BA.3, and BA.4/BA.5 variants (Figure 6 and Figure S5).

With regard to the increased breadth of response in the youngest cohort, a lot of discussion is dedicated to an original antigenic sin hypothesis, however additional references could be made to findings in adults that increased response breadth is also similarly apparent after 3x vs 2x vaccine doses. It would be interesting to compare the breadth of response seen in the youngest cohort after receiving only 2 doses of the vaccine regimen, which would help distinguish between the two hypotheses of number of doses vs effect of previous exposures. Similarly, it would be interesting to compare how the breadth of reactivity in the 3x vaccinated children compares to the breadth seen in adults after 3 doses, although I appreciate this may be outside the scope of this particular study.

Overall, given findings in adults that vaccine dose is related to increasing response magnitude and that a 3rd dose may be related to increasing response breadth, it seems likely that these factors alone may influence the differences seen in the different vaccination age groups. The authors acknowledge this to some degree but it seems like an important observation to make since if this is indeed the main driver of the age-group differences seen does this imply that utilising larger doses for younger age groups and an additional dose for older age-groups would help broaden protection against newly emerging variants?

Response: We have added additional references and discussed the findings further in discussion section.

Lines 281-297: Following mRNA vaccination, we observed higher homologous WA-1 neutralization titers in older children compared with younger children. In contrast, the breadth of the response to SARS-CoV-2 variants relative to WA.1 reactivity, appears to be highest in the youngest age group (<5 years) compared with older children. Our study findings suggest the quality of antibody response induced following SARS-CoV-2 vaccination in young children may be less vulnerable to mutations in spike of the emerging SARS-CoV-2 variants in contrast to older children. One possible explanation for these qualitative antibody differences between age groups could be due to three doses of the 3 mcg mRNA vaccine in the primary vaccination schedule for these youngest children (<5 years) compared with two vaccine doses (10 or 30 mcg mRNA) for the older children (>5 years). Clinical studies in young children <5 years showed that three vaccinations with low mRNA dose (3 mcg mRNA) were required to generate equivalent neutralizing antibody response against the vaccine-matched ancestral WA-1 strain, as was induced by two doses (30 mcg mRNA) in older children ¹⁷. It is possible that a third vaccine dose in children can improve the breadth of immune response against variants similar to the observations of broader cross-reactivity to Omicron variants after third versus second vaccine doses in adults ¹⁸. In future studies, we plan to investigate the impact of the vaccine doses (three versus two) on the breadth of reactivity in an age stratified child cohort.

Lines 330-339: The data suggests that an additional vaccine dose of the new bivalent vaccine formulation would help broaden protection against newly emerging SARS-CoV-2 variants in children. Even children with prior SARS-CoV-2 infection will immensely benefit from hybrid immunity following vaccination to efficiently protect against circulating and emerging SARS-CoV-2 variants. Importantly, with over 3.6 million infants born in the U.S each year, the population of COVID-19 naïve individuals will continue to grow. Vaccination remains an important, effective strategy in protecting against severe COVID-19 and MIS-C. This study demonstrates that effective vaccine approaches are required to generate more broadly durable protective immunity against COVID-19 or MIS-C caused by circulating Omicron BQ.1, BQ.1.1 and XBB.1 subvariants, which could influence the transmission and clinical outcome in the pediatric population.

Minor comments:

Line 111:

"Acute children <5 years old also required significantly less respiratory support than adolescents with the same acute illness (p=0.02). There were no differences in disease severity based on ICU admission, respiratory support, or mechanical ventilation among different MIS-C age groups."

More accurate to say "no significant differences found", also it is not clear that any attempt was made to account for multiple testing using Pearson's chi-squared test.

Response: Multiple testing was accounted for using Pearson's chi-squared test.

The statement (lines 129-131) was revised to: Among different MIS-C age groups, no significant differences were observed in disease severity based on ICU admission, respiratory support, or mechanical ventilation.

This sentence on line 147 is very difficult to understand:

"The seropositivity rate against WA-1 (PsVNA50 titer of >1:40) predicted to provide protection against severe COVID-19 ¹⁶ increased from 30% for acute COVID-19 patients to 95-100% for the convalescent COVID-19 for the MIS-C samples in the youngest age cohort (Figure S1a)."

Response (lines 165-8): The statement was revised to: In the youngest age cohort, the seropositivity rate against WA-1 (PsVNA50 titer of >1:40) predicted to provide protection against severe COVID-19 ¹⁶ increased from 30% for acute COVID-19 patients to 95-100% for the convalescent COVID-19 children and the MIS-C patients (Figure S1a).

Line 44:

"Monovalent mRNA vaccination induced more cross-neutralizing antibodies against SARS-CoV-2 variants especially BQ.1, BQ.1.1 and XBB.1 in young children <5 years compared with ≥5 years old children." More cross-neutralizing compared to what?

Response (lines 44-46): The statement was revised to: In contrast, monovalent mRNA vaccination induced more cross-neutralizing antibodies in young children <5 years against SARS-CoV-2 variants especially BQ.1, BQ.1.1 and XBB.1 compared with ≥5 years old children.

Line 43:

"In contrast, BQ.1, BQ.1.1 and XBB.1 subvariants are more resistant".

Were not so well neutralized?

Response (lines 43-44): The statement was revised to: The post-infection pediatric samples did not neutralize BQ.1, BQ.1.1 and XBB.1 subvariants.

Line 55:

"widespread COVID-19 around the globe"

circulation?

Response (lines 53-55): The statement was revised to: The SARS-CoV-2 Omicron variants continue to evolve, generating multiple sub-lineages with increased transmissibility and antibody-escape mutations resulting in widespread circulation of COVID-19 around the globe ¹.

Line 265:

"Children remain at the epicenter of viral transmission, therefore, induction of broadly cross-neutralizing antibodies by current and new vaccination approaches can reduce/block the transmission cycles to limit further evolution of new more transmissible and antibody-resistant SARS-CoV-2 variants."

It is also possible that this could drive further/quicker antigenic evolution through increased immune pressure in this population.

Response: The statement was revised to (lines 322-327): Children remain at the epicenter of viral transmission, and therefore induction of broadly cross-neutralizing antibodies by current and new vaccination approaches that can reduce/block the transmission cycles are required to limit further evolution of new more transmissible and antibody-resistant SARS-CoV-2 variants. It is plausible that a suboptimal neutralizing antibody response against circulating SARS-CoV-2 variants could potentially drive antigenic evolution through increased immune pressure in this population.

Reviewer #2 (Remarks to the Author):

This study compares two groups of children; i) age-stratified children with acute, severe COVID-19 (n=151) and ii) age-stratified vaccinated (BNT162b2) children (n=62). Neutralising antibody titres were examined against Omicron variants circulating in recent months (BA.4/BA.5, BA.2.75, BQ.1, BQ.1.1 and XBB.1).

Findings:

Vaccinated adolescents had higher titres against ancestral strain than <12 years.

Infected children had low titres against BA.2., BA.3, BA.4/BA.5, and BA.2.75, and minimal activity against BQ.1, BQ.1.1 and XBB.1.

Vaccination induced higher titres against variants than infected children, with measurable, but low activity against BQ.1, BQ.1.1 and XBB.1.

As the post-infection samples were collected between April 2020 and March 2021, when the prevalent circulating lineages in the US were the ancestral strain followed by the Alpha variant, it is unclear what value can be discerned from studying cross-neutralisation of Omicron-derived lineages in this population (perhaps this could be clarified).

Response: Most children in US have not received a SARS-CoV-2 booster vaccine. The primary vaccination series was based on monovalent vaccine containing the prototype WA-1 spike. The focus of the study was to evaluate the age-stratified immune response generated following vaccination with monovalent vaccine containing the prototype WA-1 spike and therefore they were compared with matched children infected with ancestral strains initially during the pandemic, so as to have appropriate post-infection comparator with same WA-1 strain contained in the vaccine. We have clarified it further in revised manuscript

Results lines 98-105: Most children in US have not received a SARS-CoV-2 booster vaccine (Figure 1a). The main aim of the study was to evaluate the capacity of the pediatric antibody response induced by the first generation monovalent COVID-19 mRNA vaccines encoding prototype WA-1 spike antigen compared with those generated following primary SARS-CoV-2 infection by ancestral WA-1 like strains early during the pandemic to neutralize currently circulating highly transmissible Omicron subvariants, in age-stratified children, during acute infection and after convalescence. Therefore, we analyzed post-SARS-CoV-2 infection or post-vaccination samples from a total of 213 US children and adolescents (Figure 1b).

The sera collected from vaccinated children were all collected approximately 20 days post vaccination, and comparisons between the three age groups suggest an age-related increase in antibody titres. The comparison between cohorts (acute COVID-19, MIS-C and convalescent) is difficult as acute samples were collected within 2 weeks of presentation while convalescent sera were approximately 60-70 days post positive test. Hence, the immune response in the acute cohorts will have had less time to develop and may be suppressed due to acute viral disease. Age-stratified within cohort comparisons would be valid, and an age-related increase in titres was observed in the acute cohort.

Response: We agree with the reviewer that the acute COVID-19 cohort has less time to develop antibodies. The convalescent cohort who had mild disease and were never hospitalized or the MIS-C who are estimated to be 3-6 weeks after their initial infection as it is a post-infectious complication, would be more similar to the vaccinated children. We have clarified and discussed this further.

Lines 114-124: There was no statistical difference in the timing of pediatric sample collection relative to hospital admission or PCR/antibody positive test between the various age groups in any of the disease cohorts (Table S2).

MIS-C is a hyperinflammatory syndrome that occurs approximately 3-6 weeks post-SARS-CoV-2 infection in children who were asymptomatic or had mild illness upon initial infection. As a control group for MIS-C, we collected convalescent COVID-19 samples from outpatient children and adolescents with asymptomatic or mild illness (never hospitalized) approximately 3-6 weeks after their positive SARS-CoV-2 PCR test. Because MIS-C is rare (~2-3 per 10,000 SARS-CoV-2 infections in US), and it is not possible to identify patients in advance, therefore, it is not feasible to get baseline (during initial infection) samples in most MIS-C patients.

Lines 268-273: The acute COVID-19 cohort has less time to develop antibodies and antibody response may be suppressed due to acute viral disease. While the convalescent cohort who had mild disease and were never hospitalized with acute COVID-19, or the MIS-C who are estimated to be 3-6 weeks after their initial SARS-CoV-2 infection as it is a post-infectious complication, would have sufficient time to generate anti-SARS-CoV-2 immune response.

In the convalescent cohort, titres were similar across all ages.

Antigenic cartography demonstrated that, irrespective of the cohort, the Omicron lineages and the ancestral variant fall into two distinct neutralisation groups. In the acute cohort, the data suggest low magnitude and heterogeneity of the neutralising response in the youngest group, while convalescent samples showed similar magnitude and heterogeneity across age groups. The analysis of “antigenic landscape” was not quantified, hence there is no statistical confidence in the conclusions. Could the antigenic landscape data be compared statistically?

Response: We have performed the statistical comparative analysis of the antigenic landscapes, and this is shown in new figures 5 and 6. The findings are described in the revised manuscript.

Lines 230-257: To quantify the magnitude of this antigenic relationship, we evaluated the antigenic landscape in these post-infection and post-vaccination children’s cohorts (Figure S5). An analysis of the three-dimensional antigenic landscape of three different age groups revealed interesting findings (Figure 5). As expected, in all age groups, mRNA vaccination induced higher cross-reactive neutralizing antibodies against variants than post-infection cohorts. In older children (5 years and older), the slope of the post-vaccination antigenic landscape and post-infection landscape was similar against the variants (Figure 5b-c). Interestingly, the slope of the landscape for vaccination induced antibody response in the youngest children (<5 years) was shallower compared with much steeper slope observed for the vaccinated older children (5 years and older), suggesting a broader cross-reactive neutralizing antibodies following vaccination in this youngest age cohort (Figure 5).

Within disease categories, both the magnitude of the neutralization titers and heterogeneity of antigenic diversity increases in age-dependent manner in the acute COVID-19 cohort (Figure 6). The youngest age group (<5 years) shows low magnitude and the Omicron variants clustered closer to the ancestral WA-1, while acute COVID-19 adolescents showing greater heterogeneity in both titers and distance between WA1 and Omicron subvariants (Figure S5). The convalescent COVID-19 and MIS-C samples display similar antigenic landscape across various age groups (Figure 6 and S5).

In the mRNA vaccinated children, the neutralizing antibodies induced following vaccination in the youngest age cohort even though lower in magnitude against

vaccine-homologous WA-1, are more cross-reactive against the circulating Omicron subvariants and are closer together on antigenic map (Figure 6 and Figure S5). However, with increase in age, the magnitude of vaccination-induced anti-WA-1 titer also increased, but the antibody response in older children (>5 years) tends to be significantly less cross-reactive against Omicron subvariants especially the newly emerged XBB.1, BQ.1, and BQ.1.1 subvariants compared with younger children (<5 years), as demonstrated by the steep slope and spread of the antigenic landscape of the neutralization for the different variants, with a notable distance from WA-1 and other BA.2, BA.2.75, BA.3, and BA.4/BA.5 variants (Figure 6 and Figure S5).

While the analyses of the neutralising activity and antigenic cartography are both valid and interesting, the rationale for comparing post-exposure sera from early in the pandemic, with post-vaccination sera to look at cross-neutralisation of late pandemic variants is not clear. The main conclusion is that for these cohorts, vaccination induced a more broad neutralising response than infection, further comparisons between the cohorts are likely confounded by the mis-matched dates of sampling post-exposure.

Response: As mentioned in response to an earlier comment, most children in US have not received a SARS-CoV-2 booster vaccine. The primary vaccination series was based on monovalent vaccine containing the prototype WA-1 spike. The focus of the study was to evaluate the age-stratified immune response generated following vaccination with monovalent vaccine containing the prototype WA-1 spike and therefore they were compared with matched children infected with ancestral strains initially during the pandemic, so as to have appropriate post-infection comparator with same WA-1 strain contained in the vaccine.

The Supplementary Table S2 contains additional information on the timing of sample collection post-exposure for various cohorts. It is clear in Supplementary Table S2, that for acute COVID-19 patients, the median days the sample were taken from hospital admission is 1 day higher in the older group: 1 day (1, 2) for <5 years; 1 day (1, 2.5) for 5 to <12 years; 2 days (1, 4) for 12-21 years). However, the majority of samples for all groups were taken close to admission on average.

The MIS-C samples were taken slightly longer after admission than acute COVID-19, but it is a post-infectious complication that occurs approximately 3-6 weeks post-SARS-CoV-2 infection, and we do not know exactly when they were originally infected with SARS-CoV-2. The MIS-C samples were collected within same timeframe for different age groups from hospital admission: 3 days (1.3, 6) for <5 years; 2.5 days (1, 5) for 5 to <12 years; 4 days (2, 14.3) for 12-21 years).

We do not believe these differences in timing of sample collection post-exposure explain the age-related differences in antibody response.

We have clarified and discussed this further in revised manuscript

Results lines 98-105: Most children in US have not received a SARS-CoV-2 booster vaccine (Figure 1a). The main aim of the study was to evaluate the capacity of the pediatric antibody response induced by the first generation monovalent COVID-19 mRNA vaccines encoding prototype WA-1 spike antigen compared with those generated following primary SARS-CoV-2 infection by ancestral WA-1 like strains early during the pandemic to neutralize currently circulating highly transmissible Omicron subvariants, in age-stratified children, during acute infection and after convalescence. Therefore, we analyzed post-SARS-CoV-2 infection or post-vaccination samples from a total of 213 US children and adolescents (Figure 1b).

Lines 114-124: There was no statistical difference in the timing of pediatric sample collection relative to hospital admission or PCR/antibody positive test between the various age groups in any of the disease cohorts (Table S2).

MIS-C is a hyperinflammatory syndrome that occurs approximately 3-6 weeks post-SARS-CoV-2 infection in children who were asymptomatic or had mild illness upon initial infection. As a control group for MIS-C, we collected convalescent COVID-19 samples from outpatient children and adolescents with asymptomatic or mild illness (never hospitalized) approximately 3-6 weeks after their positive SARS-CoV-2 PCR test. Because MIS-C is rare (~2-3 per 10,000 SARS-CoV-2 infections in US), and it is not possible to identify patients in advance, therefore, it is not feasible to get baseline (during initial infection) samples in most MIS-C patients.

Lines 268-273: The acute COVID-19 cohort has less time to develop antibodies and antibody response may be suppressed due to acute viral disease. While the convalescent cohort who had mild disease and were never hospitalized with acute COVID-19, or the MIS-C who are estimated to be 3-6 weeks after their initial SARS-CoV-2 infection as it is a post-infectious complication, would have sufficient time to generate anti-SARS-CoV-2 immune response.

Reviewer #3 (Remarks to the Author):

This manuscript described the neutralising antibody responses against the current circulating omicron subvariants in children and adolescents previously infected with the ancestral Wuhan or Alpha strain. The authors found that the age of children/adolescent may influence the magnitude and cross-reactivity of neutralising antibody responses following vaccination or infection. The strength of the paper is the unique samples cohort that comprises of children/adolescents samples with a range of clinical diseases and ages. I have several queries and comments on the paper:

Response: We thank the reviewer for the appreciating our study.

1. What is the context/relevance of comparing acute vs MIS-C or convalescent or vaccinated? It is understood that acute samples would generally be lower than convalescent. In addition, Line 142-143: please remove “demonstrated an increase in WA-1 neutralizing antibodies in younger children”. They are not directly comparable to acute patients since they are different individuals. It might indicate that children under 5 takes longer to mount full antibody response. Furthermore, there was no increase in antibody levels between samples from acute and convalescent adolescents, suggesting that samples collected during acute phase may be almost convalescent samples?

Response: We have modified manuscript to provide the relevance of comparing different post-infection cohorts.

In children, SARS-CoV-2 infection is often asymptomatic or causes mild disease; however, children are susceptible to develop severe manifestations of COVID-19 and its associated post-infectious severe complication MIS-C (lines 55-58). Therefore, we compared the two sever outcomes: either hospitalized acute COVID-19 or post-infectious hospitalized MIS-C, which happens 3-6 weeks post-exposure. As a control group for MIS-C, we collected convalescent COVID-19 samples from outpatient children and adolescents with asymptomatic or mild illness (never hospitalized) approximately 3-6 weeks after their positive SARS-CoV-2 PCR test. There was an increase in neutralizing antibody response from acute COVID-19 to convalescent COVID-19 or MIS-C samples from younger children as shown in supplementary figure S1a-b.

We have revised the Supplementary Table S2 and added information on the timing of sample collection for various cohorts. It is clear in Supplementary Table S2, that for acute COVID-19 patients, the median days the sample were taken from hospital admission is 1 day higher in the older group: 1 day (1, 2) for <5 years; 1 day (1, 2.5) for 5 to <12 years; 2 days (1, 4) for 12-21 years). However, the majority of samples for all groups were taken close to admission on average.

The MIS-C samples were taken slightly longer after admission than acute COVID, but it is a post-infectious complication that occurs approximately 3-6 weeks post-SARS-CoV-2 infection, and we do not know exactly when they were originally infected with SARS-CoV-2. The MIS-C samples were collected within same timeframe for different age groups from hospital admission: 3 days (1.3, 6) for <5 years; 2.5 days (1, 5) for 5 to <12 years; 4 days (2, 14.3) for 12-21 years).

We do not believe these differences in timing of sample collection explain the age-related differences in antibody response.

We have added the following information to reflect the timing of the sample in the revised manuscript.

Lines 114-124: There was no statistical difference in the timing of pediatric sample collection relative to hospital admission or PCR/antibody positive test between the various age groups in any of the disease cohorts (Table S2).

MIS-C is a hyperinflammatory syndrome that occurs approximately 3-6 weeks post-SARS-CoV-2 infection in children who were asymptomatic or had mild illness upon initial infection. As a control group for MIS-C, we collected convalescent COVID-19 samples from outpatient children and adolescents with asymptomatic or mild illness (never hospitalized) approximately 3-6 weeks after their positive SARS-CoV-2 PCR test. Because MIS-C is rare (~2-3 per 10,000 SARS-CoV-2 infections in US), and it is not possible to identify patients in advance, therefore, it is not feasible to get baseline (during initial infection) samples in most MIS-C patients.

Lines 268-273: The acute COVID-19 cohort has less time to develop antibodies and antibody response may be suppressed due to acute viral disease. While the convalescent cohort who had mild disease and were never hospitalized with acute COVID-19, or the MIS-C who are estimated to be 3-6 weeks after their initial SARS-CoV-2 infection as it is a post-infectious complication, would have sufficient time to generate anti-SARS-CoV-2 immune response.

The statement has been rectified for Lines 159-162: However, samples from children with convalescent non-hospitalized outpatient COVID-19 or inpatient hospitalized MIS-C, demonstrated higher WA-1 neutralizing antibodies in younger children such that PsVNA50 titers were similar across all age groups (Figure 2b-c).

2. How do you define MISC when acute patients also requires ICU admission and respiratory support? MISC usually occurs 2 weeks after covid-19- how does that influence the timing of blood sampling- please clarify.

Response: The disease was defined based on CDC case definitions. We have added the definition for cohorts in the method section.

We have revised the Supplementary Table S2 and added information on the timing of sample collection for various cohorts. It is clear in Supplementary Table S2, that for acute COVID-19 patients, the median days the sample were taken from hospital admission is 1 day higher in the older group: 1 day (1, 2) for <5 years; 1 day (1, 2.5) for 5 to <12 years; 2 days (1, 4) for 12-21 years). However, the majority of samples for all groups were taken close to admission on average.

The MIS-C samples were taken slightly longer after admission than acute COVID, but it is a post-infectious complication that occurs approximately 3-6 weeks post-SARS-CoV-2 infection, and we do not know exactly when they were originally infected with SARS-CoV-2. The MIS-C samples were collected within same timeframe for different age groups from hospital admission: 3 days (1.3, 6) for <5 years; 2.5 days (1, 5) for 5 to <12 years; 4 days (2, 14.3) for 12-21 years).

We do not believe these differences in timing of sample collection explain the age-related differences in antibody response.

Lines 346-361: The 151 U.S. pediatric patients were enrolled across multiple hospitals in the Overcoming COVID-19 Network comprised 3 independent cohorts: 54 hospitalized with acute COVID-19, 64 hospitalized with MIS-C, and 33 non-hospitalized children with mild SARS-CoV-2 infection with convalescent samples (≥ 30 days post-acute) enrolled as an at-risk for MIS-C control group, all divided in different age groups (Figure 1b). MIS-C and acute COVID-19 were defined using U.S. Centers for Disease Control and Prevention (CDC) case definitions (<https://wwwn.cdc.gov/nndss/conditions/coronavirus-disease-2019-covid-19/case-definition/2020/> approved April 5th, 2020 and <https://www.cdc.gov/mis-c/hcp/> published 5/14/2021). All acute and convalescent COVID-19 pediatric patients had SARS-CoV-2 detected by reverse transcriptase quantitative PCR (RT-qPCR) and MIS-C patients had positive SARS-CoV-2 antibody and/or RT-qPCR tests. Samples from the post-infection pediatric cohorts (acute COVID-19 vs convalescent COVID-19 vs MIS-C) were collected from a prospectively enrolling multicenter study (Overcoming COVID-19). Children's acute COVID-19 samples (serum or plasma) were collected as early as possible during hospitalization and/or study enrollment. The demographic for the pediatric cohort is shown in Table S2. The clinical information for the post-infection pediatric cohorts were described before¹³.

Lines 114-124: There was no statistical difference in the timing of pediatric sample collection relative to hospital admission or PCR/antibody positive test between the various age groups in any of the disease cohorts (Table S2).

MIS-C is a hyperinflammatory syndrome that occurs approximately 3-6 weeks post-SARS-CoV-2 infection in children who were asymptomatic or had mild illness upon initial infection. As a control group for MIS-C, we collected convalescent COVID-19 samples from outpatient children and adolescents with asymptomatic or mild illness (never hospitalized) approximately 3-6 weeks after their positive SARS-CoV-2 PCR test. Because MIS-C is rare (~2-3 per 10,000 SARS-CoV-2 infections in US), and it is not possible to identify patients in advance, therefore, it is not feasible to get baseline (during initial infection) samples in most MIS-C patients.

3. It is a little misleading to say vaccinated children under 5 demonstrate more cross-reactivity than older children/adolescents since children under 5 years old received 3 doses while children aged ≥ 5 years old as well as adolescents up to 21 years old received 2 doses. In adults, 3 doses are needed to generate neutralising antibodies against omicron variants. Therefore, the neutralising antibody responses may not be directly comparable in terms of magnitude and cross-reactivity between the different age group who received different doses of vaccine. This may influence the results for Figure 4/antigenic landscape if they are not comparable

Response: We have discussed the nuances of the vaccination regimen in children throughout the revised manuscript.

The children received 3 doses who were <5 years but these were the smallest mRNA doses (3 mcg mRNA). Clinical studies in <5 years showed that immune response generated by 2 vaccinations with low mRNA dose (3 mcg mRNA) were insufficient to generate good immune response against the vaccine-matched homologous WA-1 strain and therefore a 3rd dose was required in <5 years old to generate similar antibody response as 2-doses (10 or 30 mcg mRNA) in older children.

We have added a new supplementary figure S2 that shows the neutralization data normalized to the WA-1 titers to clarify that youngest children demonstrate broader cross-reactive SARS-CoV-2 neutralization titers. We have added information in the results section.

Lines 196-200: When normalized to the neutralizing titers against the vaccine-homologous prototype WA-1 strain, the capacity of corresponding vaccination-induced antibodies to neutralize Omicron subvariants especially BQ.1, BQ.1.1 and XBB.1 was significantly higher for the younger children (<5 years) cohort compared with the older children (Figure S2).

This was also observed in comparative landscape analysis as shown in new figure 6.

Results Lines 241-257:

Within disease categories, both the magnitude of the neutralization titers and heterogeneity of antigenic diversity increases in age-dependent manner in the acute COVID-19 cohort (Figure 6). The youngest age group (<5 years) shows low magnitude and the Omicron variants clustered closer to the ancestral WA-1, while acute COVID-19 adolescents showing greater heterogeneity in both titers and distance between WA1 and Omicron subvariants (Figure S5). The convalescent COVID-19 and MIS-C samples display similar antigenic landscape across various age groups (Figure 6 and S5).

In the mRNA vaccinated children, the neutralizing antibodies induced following vaccination in the youngest age cohort even though lower in magnitude against vaccine-homologous WA-1, are more cross-reactive against the circulating Omicron subvariants and are closer together on antigenic map (Figure 6 and Figure S5). However, with increase in age, the magnitude of vaccination-induced anti-WA-1 titer also increased, but the antibody response in older children (>5 years) tends to be significantly less cross-reactive against Omicron subvariants especially the newly emerged XBB.1, BQ.1, and BQ.1.1 subvariants compared with younger children (<5 years), as demonstrated by the steep slope and spread of the antigenic landscape of the neutralization for the different variants, with a notable distance from WA-1 and other BA.2, BA.2.75, BA.3, and BA.4/BA.5 variants (Figure 6 and Figure S5).

We have added additional references and discussed the findings further in discussion section.

Lines 281-297: Following mRNA vaccination, we observed higher homologous WA-1 neutralization titers in older children compared with younger children. In contrast, the breadth of the response to SARS-CoV-2 variants relative to WA.1 reactivity, appears to be highest in the youngest age group (<5 years) compared with older children. Our study findings suggest the quality of antibody response induced following SARS-CoV-2 vaccination in young children may be less vulnerable to mutations in spike of the emerging SARS-CoV-2 variants in contrast to older children. One possible explanation for these qualitative antibody differences between age groups could be due to three doses of the 3 mcg mRNA vaccine in the primary vaccination schedule for these youngest children (<5 years) compared with two vaccine doses (10 or 30 mcg mRNA) for the older children (>5 years). Clinical studies in young children <5 years showed that three vaccinations with low mRNA dose (3 mcg mRNA) were required to generate equivalent neutralizing antibody response against the vaccine-matched ancestral WA-1 strain, as was induced by two doses (30 mcg mRNA) in older children¹⁷. It is possible that a third vaccine dose in children can improve the breadth of immune response against variants similar to the observations of broader cross-reactivity to Omicron variants after third versus second vaccine doses in adults¹⁸. In future studies, we plan to investigate the impact of the vaccine doses (three versus two) on the breadth of reactivity in an age stratified child cohort.

Lines 330-339: The data suggests that an additional vaccine dose of the new bivalent vaccine formulation would help broaden protection against newly emerging SARS-CoV-2 variants in children. Even children with prior SARS-CoV-2 infection will immensely benefit from hybrid immunity following vaccination to efficiently protect against circulating and emerging SARS-CoV-2 variants. Importantly, with over 3.6 million infants born in the U.S each year, the population of COVID-19 naïve individuals will continue to grow. Vaccination remains an important, effective strategy in protecting against severe COVID-19 and MIS-C. This study demonstrates that effective vaccine approaches are required to generate more broadly durable protective immunity against COVID-19 or MIS-C caused by circulating Omicron BQ.1, BQ.1.1 and XBB.1 subvariants, which could influence the transmission and clinical outcome in the pediatric population.

4. Discussion of antigenic sin was good but other parts needs improvement. E.g. Line 268-270 is very vague. Please provide details how your findings can provide guidance which leads to my next point of study limitations- please include. The conclusion do not align with the data. E.g. what effective vaccine approaches can be derived from this study? This study applies to alpha- and/or wuhan-related infection. Hybrid immunity and immunity from omicron-infected children may be different.

Response: We have expanded the discussion.

Lines 327-339: Our data can help provide guidance on when children should be vaccinated and in designing policies for vaccination based on the immune status in settings where the risk of transmission is high, such as high-occupancy indoor

setting in schools and daycare. The data suggests that an additional vaccine dose of the new bivalent vaccine formulation would help broaden protection against newly emerging SARS-CoV-2 variants in children. Even children with prior SARS-CoV-2 infection will immensely benefit from hybrid immunity following vaccination to efficiently protect against circulating and emerging SARS-CoV-2 variants. Importantly, with over 3.6 million infants born in the U.S each year, the population of COVID-19 naïve individuals will continue to grow. Vaccination remains an important, effective strategy in protecting against severe COVID-19 and MIS-C. This study demonstrates that effective vaccine approaches are required to generate more broadly durable protective immunity against COVID-19 or MIS-C caused by circulating Omicron BQ.1, BQ.1.1 and XBB.1 subvariants, which could influence the transmission and clinical outcome in the pediatric population.

We have added the limitation to evaluate vaccine boosters containing the omicron variant spike and hybrid immunity.

Lines 295-297: In future studies, we plan to investigate the impact of the vaccine doses (three versus two) and hybrid immunity on the breadth of reactivity in an age stratified child cohort.

5. Line 147-150: “The seropositivity rate against WA-1 (PsVNA50 titer of >1:40) predicted to provide protection against severe COVID-19 increased from 30% for acute COVID-19 patients to 95-100% for the convalescent COVID-19 for the MIS-C samples in the youngest age cohort (Figure S1a)”. I would suggest to remove or reword these sentences. The reference correlate is for vaccination and may not applies to all variants.

Response: We have clarified the statement.

Lines 165-170: In the youngest age cohort, the seropositivity rate against WA-1 (PsVNA50 titer of >1:40) predicted to provide protection against severe COVID-19 16 increased from 30% for acute COVID-19 patients to 95-100% for the convalescent COVID-19 children and the MIS-C patients (Figure S1a). However, the caveat to this correlate of protection that it was defined in the context of vaccination and may not apply to all SARS-CoV-2 variants.

6. Neutralising antibodies against the omicron subvariants in Figure 2 is hard to visualise/interpret with text. Assuming GMT10 is the minimum detection limit and anything under GMT40 is seronegative. The text should mention seronegativity/seropositivity rather than GMT10-12 or 10-22, which is negative. Please consider revising the text for results (line 171-line 187). I would also suggest using seropositivity rather than protective titers.

Response: Figure 2 has been revised to show the percent seropositivity. The text has been modified accordingly to show seropositive values.

Lines 183-203: Neutralization titers against Omicron subvariants were rare for post-infection pediatric cohorts that were infected with the ancestral WA-1 or the Alpha strain (samples collected prior to March 2021) possibly due to the presence of low cross-reactive antibodies against circulating Omicron subvariants BA.2, BA.2.75, BA.3 and BA.4/BA.5 with seropositivity rate (PsVNA50 titer of >1:40) of 0-29% across all age groups and disease cohorts (Figure 2a-c). Moreover, neutralizing antibody titers against the recently emerged highly transmissible BQ.1, BQ.1.1, and XBB.1 subvariants were further reduced with only 0-10% seropositivity across all post-infection cohorts (Figure 2a-c).

In contrast to age-dependent differences observed in neutralization titers against WA-1, the primary monovalent mRNA vaccination series in children induced similar cross-neutralizing antibody titers against the Omicron variants BA.2, BA.3 and BA.2.75 across all age groups (Figure 2d). Surprisingly, younger children (<5 years old) demonstrated higher neutralizing antibody titers against newly emerged BQ.1 (GMT:62), BQ.1.1 (GMT:54), and XBB.1 (GMT:28) subvariants compared with older children (>5 years old) with GMT of only 11-21. When normalized to the neutralizing titers against the vaccine-homologous prototype WA-1 strain, the capacity of corresponding vaccination-induced antibodies to neutralize Omicron subvariants especially BQ.1, BQ.1.1 and XBB.1 was significantly higher for the younger children (<5 years) cohort compared with the older children (Figure S2). The younger (<5 years) vaccinated children demonstrated seropositive titers (PsVNA50 titer of >1:40) against these highly transmissible BQ.1 (60%), BQ.1.1 (50%), and XBB.1 (20%) subvariants compared with only 5-10% of the vaccinated adolescents (Figure 2d).

7. Please consider rearranging results presentation- there are reference to figure 2 and then to figure 3 and then back to figure 2. Perhaps the RBD binding data can be presented first to improve readability or move to supplementary.

Response: The results for RBD binding have been rearranged to depict after describing the Figure 2 results.

Minor comments

Line 43-44: please consider rewording.

Response: Reworded to: The post-infection pediatric samples did not neutralize BQ.1, BQ.1.1 and XBB.1 subvariants.

Line 71 and line 263: would that be a reference

Response: The link is more appropriate. Index referencing would not be suitable for this CDC website link.

Line 72: please provide some figures for context.

Response: Figure 1a has been added.

Line 84: edits required

Response: Edited to: Therefore, it is critical to understand the capacity of the antibody response induced by SARS-CoV-2 vaccination compared with post-infection immune response in children of different age groups to neutralize currently circulating highly transmissible SARS-CoV-2 Omicron BA.4, BA.5, BA.2.75, BQ.1, BQ.1.1 and XBB.1 subvariants.

Results: Please provide median age for each age group, and actual number and percentage, e.g. X/X (X%) for line 108-113.

Response: We have revised the Supplementary Table S2 and added information on the age distribution (median and IQR) for various cohorts.

We have referred to the table for demographics distribution where it is clearer, rather than providing a very descriptive statements in results.

Lines 105-106: The demographics of the study cohorts are summarized in Table S2.

Line 84-88: rewording required.

Response: Reworded to (lines 85-88): Therefore, it is critical to understand the capacity of the antibody response induced by SARS-CoV-2 vaccination compared with post-infection immune response in children of different age groups to neutralize currently circulating highly transmissible SARS-CoV-2 Omicron BA.4, BA.5, BA.2.75, BQ.1, BQ.1.1 and XBB.1 subvariants.

Line 223-228: long sentence

Response: Broken into two sentences (lines 260-266): There are different COVID-19 vaccine doses and regimens for various age groups of children and lower vaccination uptake in children especially in children who had prior SARS-CoV-2 infection. Therefore, it is critical to understand the capacity of post-infection and post-vaccination antibodies to neutralize currently circulating highly contagious SARS-CoV-2 BQ.1, BQ.1.1, and XBB.1 subvariants in an age-stratified manner and across disease spectrums in children, which can help refine the vaccine approach in this important but critically under-represented and under-studied population.

Line 231: please indicate wuhan or alpha infection-induced antibodies.

Response: Added in Lines 273-276: We determined that ancestral WA-1 or Alpha infection-induced antibodies are not sufficient to neutralize highly contagious BQ.1, BQ.1.1, and XBB.1 subvariants across disease cohorts in any age group of children, and therefore vaccination would be key to prevent children against disease caused by the emerging SARS-CoV-2 variants.

Line 323 and 325: which recombinant RBD protein was used- sinobiological or in-house made.

Response: RBD used in this study is from Sino biological. Clarified in lines 407-408: Purified recombinant SARS-CoV-2 spike receptor binding domain (RBD) of WA-1 strain expressed in HEK293 Cells was purchased from Sino Biologicals (wtRBD; 40592-V08H).

REVIEWER COMMENTS

Reviewer #1 (Remarks to the Author):

The authors have made useful additional contributions to the manuscript with this revision, and in general I think it is valuable and informative work, however I still have some major concerns with the extent to which the researchers confidently link neutralization differences in the cohorts to age differences.

With regards to whether age-cohort differences are related to age itself or simply differences in vaccine dosage and number, the authors make several improvements, including a new section in the discussion relating to the possible dose-magnitude and number effects. I am also excited to see the proposed study investigating the impact of the number of vaccine doses at some point.

However, the authors are still quite loose in other sections of the text with regard to whether they are speculating that age itself is the important factor or if they think other factors are at play. For example in the abstract, line 39 they still state:

“An age-dependent effect on neutralizing antibodies was observed against 40 SARS-CoV-2 and its variants.”

Are they speculating age-dependency following vaccination, or in the acute / mild convalescent / MIS-C group?

Similarly, on line 191 they still state:

“In contrast to age-dependent differences observed in neutralization titers against WA-1”

Which seems to imply they think cohort vaccination differences are age-related, rather than dose or number of doses related. Across the paper I think it is important to be clearer where an age-related effect is hypothesised, or if it is simply differences between the age-group cohorts that could be reasonably attributed to other factors, since this is a key part of this investigation.

As mentioned, I think the discussion has been improved by the addition on lines 290-297, discussing the potential dosage and number of doses as confounders that may make vaccine differences between cohorts not in fact age-dependent. However, I still find the discussion relating to the original antigenic sin hypothesis problematic. Essentially they seem to be hypothesising that the response in younger children covers a more diverse set of conserved spike regions since the repertoire is not constrained to those that correspond to previous hCoV exposures. While this is plausible, it does not fit the literature determining the substitutions that are important for neutralization escape in adults, which are not those that are conserved with hCoV viruses. The authors also offer no explanation as to why these effects would be observed for vaccination alone and not seen in the different post-infection age-cohorts. They also now end this section with the statement:

“In support of this hypothesis, we observed that vaccination induced antibodies in younger children (<5 years) are more cross-reactive and effective in neutralizing Omicron BQ.1, BQ.1.1 and XBB.1 subvariants when normalized to the corresponding vaccine-homologous prototype WA-1 strain activity, compared with the older children (> 5 years).”

Although true, based on their useful addition of figure S2, this is not specifically evidence for an original antigenic sin based hypothesis but could equally apply to the dosage and dose number related differences.

The other key area where age-dependent effects are hypothesised is in the increased titers seen in the 12-21 year olds in the acute COVID-19 cohort. The additions made in table S2 comparing the timing of sample collection in the different age groups was helpful to begin addressing a potential confounder in this group, although it still seems unclear whether even the small differences in collection time they acknowledge play a role. In the rebuttal they assert “We do not believe these differences in timing of sample collection explain the age-related differences in antibody response.”, but this should be easy to investigate within the 12-21 year old cohort to see for example whether samples that were collected later tended to have higher titers or not?

As mentioned, the key finding seems to be that titers in the acute group are higher for the 12-21 age group in particular. If indeed related to their age this is an interesting finding but still not much discussion is given to it other than:

“In this pediatric study, we observed an age-dependent effect in antibody response to SARS-CoV-2 in acute severe infection, as compared with convalescent mild COVID-19, MIS-C, or in vaccinated children. The acute COVID-19 cohort has less time to develop antibodies and antibody response may be suppressed due to acute viral disease. While the convalescent cohort who had mild disease and were never hospitalized with acute COVID-19, or the MIS-C who are estimated to be 3-6 weeks after their

initial SARS-CoV-2 infection as it is a post-infectious complication, would have sufficient time to generate anti-SARS-CoV-2 immune response.”

Are the authors hypothesizing that the age groups <5 and 5-11 are slower to mount their initial response? If so is this a new finding or is there any prior evidence either for or against this in other studies across other pathogens for example? There are also perhaps other hypotheses to consider relating to how younger age-groups may present differently, perhaps as “acute” cases even at lower viral doses and consequently smaller antibody responses.

Finally, on closer inspection of figures 5&6 it is not clear why the points representing the mean neutralization titers do not align with the antibody landscapes in several cases. Do the authors have an explanation for what is leading to that discrepancy? Sharing the code used to generate those figures would also be helpful.

Responses to the original minor comments were well resolved by the authors.

Additional minor comments the authors may find helpful:

- Lines 81-85: could be split into two sentences?

- It looks like some of the antibody landscape figures have been stretched in an image post-processing step in order to stretch the z-axis scale. The authors may find it useful to know that there is a plotting parameter in the ablandscapes package “aspect.z” that can be set to achieve the same effect.

- Lines 114-124: “There was no statistical difference in the timing of pediatric sample collection relative to hospital admission or PCR/antibody positive test between the various age groups in any of the disease cohorts (Table S2).”

Could the authors specify what statistical test was applied?

- Lines 226-268: “In this pediatric study, we observed an age-dependent effect in antibody response to SARS-CoV-2 in acute severe infection, as compared with convalescent mild COVID-19, MIS-C, or in vaccinated children.”

Perhaps clarity could be improved with something like “In this pediatric study, we observed a potentially age-dependent effect in antibody response to SARS-CoV-2 in acute severe infection, but not for convalescent mild COVID-19 or MIS-C”. It is also not quite clear how to parse what they mean about vaccinated children in this sentence.

Reviewer #2 (Remarks to the Author):

I am happy that the authors have addressed my previous comment to the best of their ability. The potential flaws in comparing between groups have been stated in the revised text and the interpretation of the data tempered accordingly.

REVIEWERS' COMMENTS:

Reviewer #1 (Remarks to the Author):

The authors have made useful additional contributions to the manuscript with this revision, and in general I think it is valuable and informative work, however I still have some major concerns with the extent to which the researchers confidently link neutralization differences in the cohorts to age differences.

With regards to whether age-cohort differences are related to age itself or simply differences in vaccine dosage and number, the authors make several improvements, including a new section in the discussion relating to the possible dose-magnitude and number effects. I am also excited to see the proposed study investigating the impact of the number of vaccine doses at some point.

However, the authors are still quite loose in other sections of the text with regard to whether they are speculating that age itself is the important factor or if they think other factors are at play. For example in the abstract, line 39 they still state:

“An age-dependent effect on neutralizing antibodies was observed against 40 SARS-CoV-2 and its variants.”

Are they speculating age-dependency following vaccination, or in the acute / mild convalescent / MIS-C group?

Similarly, on line 191 they still state:

“In contrast to age-dependent differences observed in neutralization titers against WA-1”

Which seems to imply they think cohort vaccination differences are age-related, rather than dose or number of doses related. Across the paper I think it is important to be clearer where an age-related effect is hypothesised, or if it is simply differences between the age-group cohorts that could be reasonably attributed to other factors, since this is a key part of this investigation.

Response: We have replaced the term ‘age-dependent’ with ‘age-associated’ throughout the manuscript. Since age-associated does not infer cause and effect of age and is more accurate as other things than age itself can cause this association of the observed differences in immune response in these various age groups of children.

Lines 39-40: An age-associated effect on neutralizing antibodies was observed against SARS-CoV-2 and its variants following acute COVID-19 or vaccination.

Lines 192-195: In contrast to age-associated differences observed in neutralization titers against WA-1 that may be due to either vaccine dosage or the number of vaccine doses, the primary monovalent mRNA vaccination series in children induced similar cross-neutralizing antibody titers against the Omicron variants BA.2, BA.3 and BA.2.75 across all age groups (Figure 2d).

Lines 267-269: In this pediatric study, we observed an age-associated effect in antibody response to SARS-CoV-2 in acute severe infection or vaccination, but not for convalescent mild COVID-19 or MIS-C.

As mentioned, I think the discussion has been improved by the addition on lines 290-297, discussing the potential dosage and number of doses as confounders that may make vaccine differences between cohorts not in fact age-dependent. However, I still find the discussion relating to the original antigenic sin hypothesis problematic. Essentially they seem to be hypothesising that the response in younger children covers a more diverse set of conserved spike regions since the repertoire is not constrained to those that correspond to previous hCoV exposures. While this is plausible, it does not fit the literature determining the substitutions that are important for neutralization escape in adults, which are not those that are conserved with hCoV viruses. The authors also offer no explanation as to why these effects would be observed for vaccination alone and not seen in the different post-infection age-cohorts. They also now end this section with the statement:

“In support of this hypothesis, we observed that vaccination induced antibodies in younger children (<5 years) are more cross-reactive and effective in neutralizing Omicron BQ.1, BQ.1.1 and XBB.1 subvariants when normalized to the corresponding vaccine-homologous prototype WA-1 strain activity, compared with the older children (> 5 years).”

Although true, based on their useful addition of figure S2, this is not specifically evidence for an original antigenic sin based hypothesis but could equally apply to the dosage and dose number related differences.

Response:

We acknowledge that multiple attributes, including antigenic sin, vaccine dosage, number of vaccine doses, and the development stage of immune system, can all contribute to the age-associated differences observed following SARS-CoV-2 mRNA vaccination in these cohorts of children. We have discussed all these possibilities in the discussion section. We have revised the discussion to clarify further.

Moreover, we have mentioned that most of these cross-reactive antibodies to seasonal hCoVs target the conserved S2 domain and they do not neutralize SARS-CoV-2, as was observed for children with hCoV infection (ref 13). Majority of the SARS-CoV-2 neutralizing antibody epitopes target the S1 domain of the SARS-CoV-2 spike that contains the receptor binding domain, which is not conserved between SARS-CoV-2 and hCoVs. While we cannot compare the effect of dosage or numbers of doses in the current study, we have observed similar findings in different age group of children in a pandemic influenza vaccination study. In the H1N1pdm09 vaccination study, all cohorts got 2 doses of some vaccine dosage either with or without MF59 adjuvant. We observed similar age-dependent effect, with the youngest children showing the most diverse antibody repertoire compared

with older children and adults. We have discussed it further in the discussion section.

Lines 282-326:

Following mRNA vaccination, we observed higher homologous WA-1 neutralization titers in older children compared with younger children. In contrast, the breadth of the neutralization response to SARS-CoV-2 variants relative to WA.1 reactivity, appears to be highest in the youngest age group (<5 years) compared with older children. Our study findings suggest the quality of antibody response induced following SARS-CoV-2 vaccination in young children may be less vulnerable to mutations in spike of the emerging SARS-CoV-2 variants in contrast to older children. One possible explanation for these qualitative antibody differences between age groups could be due to three doses of the 3 mcg mRNA vaccine in the primary vaccination schedule for these youngest children (<5 years) compared with two vaccine doses of higher dosage (10 or 30 mcg mRNA) for the older children (>5 years). Clinical studies in young children <5 years showed that three vaccinations with low mRNA dose (3 mcg mRNA) were required to generate equivalent neutralizing antibody response against the vaccine-matched ancestral WA-1 strain, as was induced by two doses (30 mcg mRNA) in older children 17. It is possible that a third vaccine dose in children can improve the breadth of immune response against variants similar to the observations of broader cross-reactivity to Omicron variants after third versus second vaccine doses in adults 18. In future studies, we plan to investigate the impact of the vaccine doses (three versus two) and hybrid immunity on the breadth of reactivity in an age stratified child cohort.

The other possible mechanism can be due to the antigenic sin hypothesis, whereby older children due to their prior exposure to seasonal human coronaviruses (hCoVs) have long-lived memory B cells specific for conserved sites on the coronavirus spike, especially in SARS-CoV-2 spike S2 domain (ref 19, 20, 21, 22). Following vaccination, these memory B cells are recalled, thereby focusing the antibody response to these conserved sites in spike. Most of these cross-reactive antibodies to seasonal CoVs do not neutralize SARS-CoV-2, as was observed for children with hCoV infection (ref 13). Majority of the SARS-CoV-2 neutralizing antibody epitopes target the S1 domain of the SARS-CoV-2 spike that contains the receptor binding domain. However, since younger children (<5 years) have a shorter history of exposure to seasonal hCoVs, following vaccination they potentially could generate a more diverse antibody repertoire than immune-experienced older children, leading to higher cross-neutralization and potential efficacy against emerging SARS-CoV-2 highly contagious BQ.1, BQ.1.1 and XBB.1 subvariants. In support of this hypothesis, we observed that vaccination induced antibodies in younger children (<5 years) are more cross-reactive and effective in neutralizing Omicron BQ.1, BQ.1.1 and XBB.1 subvariants when normalized to the corresponding vaccine-homologous prototype WA-1 strain activity, compared with the older children (> 5 years) (Figure S2). Similar findings were observed of pandemic influenza vaccine studies in different age groups of children and adults who were given two doses of same vaccine dosage with or without MF59 adjuvant,

where older children or adults contain pre-existing immunity against seasonal influenza strains. Following two doses of MF59-adjuvanted 7.5 mcg H1N1pdm09 vaccine, a higher diversity and affinity maturation of antibody repertoire was observed in younger children (12-35 months) against the HA1 domain of H1N1pdm09 compared with older children (3-8 years) or adults (18-60 years) (ref 23). This study suggested an impact of antigenic sin hypothesis of non-neutralizing antibody cross-reactivity to conserved epitopes in HA2 domain on the development of novel anti-H1N1pdm09 HA1 specific immune response following vaccination. In the current study, multiple attributes, including antigenic sin, vaccine dosage, number of vaccine doses, and the development stage of immune system, can all contribute to the age-associated differences observed following SARS-CoV-2 mRNA vaccination in these cohorts of children.

The other key area where age-dependent effects are hypothesised is in the increased titers seen in the 12-21 year olds in the acute COVID-19 cohort. The additions made in table S2 comparing the timing of sample collection in the different age groups was helpful to begin addressing a potential confounder in this group, although it still seems unclear whether even the small differences in collection time they acknowledge play a role. In the rebuttal they assert “We do not believe these differences in timing of sample collection explain the age-related differences in antibody response.”, but this should be easy to investigate within the 12-21 year old cohort to see for example whether samples that were collected later tended to have higher titers or not?

Response: For the 12-21 year old acute COVID-19 cohort, we did not observe any time-point pattern or differences in the antibody response based on distribution of timing, since all samples were collected very close at hospitalization.

As mentioned, the key finding seems to be that titers in the acute group are higher for the 12-21 age group in particular. If indeed related to their age this is an interesting finding but still not much discussion is given to it other than:

“In this pediatric study, we observed an age-dependent effect in antibody response to SARS-CoV-2 in acute severe infection, as compared with convalescent mild COVID-19, MIS-C, or in vaccinated children. The acute COVID-19 cohort has less time to develop antibodies and antibody response may be suppressed due to acute viral disease. While the convalescent cohort who had mild disease and were never hospitalized with acute COVID-19, or the MIS-C who are estimated to be 3-6 weeks after their initial SARS-CoV-2 infection as it is a post-infectious complication, would have sufficient time to generate anti-SARS-CoV-2 immune response.”

Are the authors hypothesizing that the age groups <5 and 5-11 are slower to mount their initial response? If so is this a new finding or is there any prior evidence either for or against this in other studies across other pathogens for example? There are also perhaps other hypotheses to consider relating to how younger age-groups may present differently, perhaps as “acute” cases even at lower viral doses and consequently smaller antibody responses.

Response: The reviewer is correct. We have further expanded the discussion for these findings in the acute infection cohort.

Lines 327-340: The presence of pre-existing immunity to seasonal CoVs, viral kinetics, as well as the developmental stage of the immune system in these various age children's cohorts may contribute to kinetics and magnitude of immune response following acute SARS-CoV-2 infection. This age-dependent phenomenon was observed at early time-point following infection in acute COVID-19 cohort, where older children developed faster and higher neutralizing antibody response than younger children against the infecting WA-1 strain. At later time-point post-exposure (convalescent COVID-19 and MIS-C), the kinetics of antibody response in the younger children reaches similar levels as the neutralization titers of older children. Moreover, the infection induced antibody response in these three age groups at 3 to 6-weeks post-infection (convalescent COVID-19 and MIS-C), comparable to the 4-week time-point post-vaccination, showed similar neutralization antibody response against the infecting ancestral WA-1 strain as well as Omicron subvariants. These differences observed between infection vs vaccination induced breadth of immune response to Omicron subvariants in various age groups of children, and mechanism underlying this antibody repertoire, remain an area of intense investigation.

Finally, on closer inspection of figures 5&6 it is not clear why the points representing the mean neutralization titers do not align with the antibody landscapes in several cases. Do the authors have an explanation for what is leading to that discrepancy? Sharing the code used to generate those figures would also be helpful.

Response: The antibody landscapes were generated based on the revised code that was referred and published by Smith and Fonville's groups (Ref 29-31), as indicated in the methods section. We have provided the code for the figures 5 and 6 in this revised manuscript.

The neutralizing antibody landscapes shown for the different pediatric groups in Figures 5 and 6 were constructed from antibody landscapes for each individual sample of respective group. The overall neutralizing antibody landscape is based on the distribution of the three-dimensional antibody landscape of these individual serum in that pediatric group across the various SARS-CoV-2 strains shown on the x-y base plane. The gray impulses with colored symbols on z-axis show the height of the GMT for a specific variant and are identical to the GMT shown in figure 2.

The x-y plane is given by the antigenic cartography map, while the height of the landscape over a particular point or variant represents the estimated magnitude of serum reactivity in that antigenic region. Antibody landscapes therefore give an indication of how pediatric sample reactivity distributes after exposure to different variants, how the magnitudes of the responses compare, and predicts expected

levels of reactivity to variants that have not been titrated for other possible variants that may be present in the x-y antigenic space.

In revised figures 5 and 6, for clarification, we have enlarged the circle symbols and colored them to represent the corresponding GMT for that group. We have also revised text and figure legends with this information.

Lines 232-234: An analysis of the three-dimensional antigenic landscape of three different age groups, corresponding with the neutralization titers presented in Figure 2, revealed interesting findings (Figure 5).

Figure 5. Neutralizing antibody landscapes following infection or vaccination for different age groups of children. The antigenic landscape was generated using the SARS-CoV-2 neutralization titers against WA-1 and Omicron BA.2, BA.3, BA.2.75, BA.4/BA.5, BQ.1, BQ1.1 and XBB.1 for the 213 children with either acute COVID-19, convalescent COVID-19, MIS-C or following vaccination, divided by age categories: (a) <5 years (n=65; 23 acute, 10 convalescent, 22 MIS-C and 10 naïve vaccinated), (b) 5-11 years (n=71; 10 acute, 11 convalescent, 18 MIS-C and 32 naïve vaccinated) and (c) 12-21 years old (n=77; 21 acute, 12 convalescent, 24 MIS-C and 20 naïve vaccinated). The base x and y axis of each landscape map represents antigenic cartography between WA-1 and variants, with colored points representing the locations of each SARS-CoV-2 strain. The grid squares (1 antigenic unit) correspond to a 2-fold change in neutralization assay. The vertical z-axis corresponds to the neutralization titer on the log₂ scale. The overall neutralizing antibody landscape for each pediatric group was constructed by fitting individual neutralizing antibody landscapes for each children's serum sample for that group against the respective SARS-CoV-2 strain. The landscapes are color coded according to patient categories (acute, convalescent, MIS-C, and vaccinated). The magnitude of neutralization titers (PsVNA50 GMTs) for children either following vaccination or infection against each SARS-CoV-2 strain are shown by impulses connected for each SARS-CoV-2 variant on the z-axis in the landscape and the filled-circle symbols are color-coded by each patient category group. A Mann-Whitney test in R package was used to determine the significance of the difference between patient cohort for each age group, and the significant p-values are shown.

Figure 6. Comparison of age-stratified antigenic landscapes in different post-infection and post-vaccination children's cohorts. The children's cohorts were divided into four categories: acute COVID-19, convalescent COVID-19, MIS-C, and naïve vaccinated. The antigenic landscape was generated using the SARS-CoV-2 neutralization titers against WA-1 and Omicron BA.2, BA.3, BA.2.75, BA.4/BA.5, BQ.1, BQ1.1 and XBB.1 for the 213 children with either acute COVID-19 (n=54), convalescent COVID-19 (n=33), MIS-C (n=64) or following vaccination (n=62) in different age groups. The landscapes are color coded according to age-stratified children. The x and y axis of each landscape map represent 2D antigenic

cartography between WA-1 and variants, with colored points representing the locations of each SARS-CoV-2 strain. The grid squares (1 antigenic unit) represent antigenic distance that corresponds to a 2-fold change in neutralization titer. The vertical z-axis corresponds to the neutralization titer on the log₂ scale. The overall neutralizing antibody landscape for each cohort was constructed by fitting individual neutralization landscapes for each pediatric serum sample for that age group against the respective SARS-CoV-2 strain. The magnitude of neutralization titers (PsVNA50 GMTs) for different age groups against each SARS-CoV-2 variant are shown by impulses connected for each variant on the z-axis in the landscape and the filled-circle symbols are color-coded by each age group. A Mann-Whitney test in R package was used to determine the significance of the difference between children age groups for each patient cohort, and the significant p-values are shown.

Responses to the original minor comments were well resolved by the authors.

Additional minor comments the authors may find helpful:

- Lines 81-85: could be split into two sentences?

Response: Done.

- It looks like some of the antibody landscape figures have been stretched in an image post-processing step in order to stretch the z-axis scale. The authors may find it useful to know that there is a plotting parameter in the ablandscapes package “aspect.z” that can be set to achieve the same effect.

Response: Thanks for the suggestion. All figures are fixed.

- Lines 114-124: “There was no statistical difference in the timing of pediatric sample collection relative to hospital admission or PCR/antibody positive test between the various age groups in any of the disease cohorts (Table S2).”

Could the authors specify what statistical test was applied?

Response: When comparing the three age groups within each disease category using the Kruskal-Wallis test, there were no statistical differences in the timing of pediatric blood sample collection relative to the PCR/antibody positive test date for acute and convalescent patients or hospital admission date for MIS-C patients. This information has been added as a footnote in Table S2.

- Lines 226-268: “In this pediatric study, we observed an age-dependent effect in antibody response to SARS-CoV-2 in acute severe infection, as compared with convalescent mild COVID-19, MIS-C, or in vaccinated children.”

Perhaps clarity could be improved with something like “In this pediatric study, we observed a potentially age-dependent effect in antibody response to SARS-CoV-2 in acute severe infection, but not for convalescent mild COVID-19 or MIS-C”. It is also not quite clear how to parse what they mean about vaccinated children in this sentence.

Response: The sentence has been clarified.

Lines 267-269: In this pediatric study, we observed an age-associated effect in antibody response to SARS-CoV-2 in acute severe infection or vaccination, but not for convalescent mild COVID-19 or MIS-C.

Reviewer #2 (Remarks to the Author):

I am happy that the authors have addressed my previous comment to the best of their ability. The potential flaws in comparing between groups have been stated in the revised text and the interpretation of the data tempered accordingly.

Response: We thank the reviewer for the positive response.